# The preference signature of the SARS-CoV-2 Nucleocapsid NTD for its 5'-genomic RNA elements

Sophie Marianne Korn [1,2], Karthikeyan Dhamotharan [1,2], Cy M. Jeffries [3] & Andreas Schlundt [1,2] ✉

The nucleocapsid protein (N) of SARS-CoV-2 plays a pivotal role during the viral life cycle. It is involved in RNA transcription and accounts for packaging of the large genome into virus particles. N manages the enigmatic balance of bulk RNA-coating versus precise RNA-binding to designated cis-regulatory elements. Numerous studies report the involvement of its disordered segments in non-selective RNA-recognition, but how N organizes the inevitable recognition of specific motifs remains unanswered. We here use NMR spectroscopy to systematically analyze the interactions of N's N-terminal RNA-binding domain (NTD) with individual cis RNA elements clustering in the SARS-CoV-2 regulatory 5'-genomic end. Supported by broad solution-based biophysical data, we unravel the NTD RNA-binding preferences in the natural genome context. We show that the domain's flexible regions read the intrinsic signature of preferred RNA elements for selective and stable complex formation within the large pool of available motifs.

Despite impressive progress in vaccination, coronaviral infections will remain a severe threat to human health. This calls for efforts towards curing infections based on a detailed understanding of the most prevalent molecular interactions specific to the virus. The responsible pathogen of the Covid-19 pandemic, the severe acute respiratory syndrome coronavirus-2 (SARS-CoV-2, SCoV-2) carries a large single-stranded (ss) RNA genome of almost 30000 nucleotides (nt)[1]. As for other CoVs, it is subject to genomic and sub-genomic transcription[2], replication[3] and condensation into newly assembled particles with each round of viral propagation[4].

For the packaging of viral RNA SCoV-2 uses its versatile nucleocapsid protein (N)[4]. N is more than 90% conserved to the SCoV homolog, appears most abundant on the general protein level[5] and in RNA-protein complexes (RNPs)[6,7]. N integrates five distinct parts (often termed N1-N5), comprising two domains that are intertwined by intrinsically disordered regions (IDR), with IDR2 (N3) embedding the SR-region (Fig. 1a). While the C-terminal domain (CTD, N4) constitutes dimerization, the NTD (N2) represents an RNA-binding domain (RBD)

with a right hand-like fold and a five-stranded, antiparallel β-sheet core (Fig. 1b). A distinct feature is the large β-hairpin (β2'- β3', residues 90–107). Its highly basic composition prompts its involvement in nucleic acid interactions[8].

N gathers multiple functions[9] in genome processing, e.g. its role in RNA synthesis[10–12], and immune evasion[13–15]. For its primary function, genome encapsidation, N combines protein oligomerization with RNA-binding[16–18], which in combination accounts for the occurrence of RNA-protein condensates both in vitro and in cells. For this, a plethora of studies since the SCoV-2 outbreak has intensively addressed the influence of N phosphorylation[16], the contributions of N domains[18,19], the role of viral RNA[20], the viral nsp3[21], and host proteins[22]. Very recent work ultimately claimed that phase separation in infected cells enhances viral transcription and assembly[16,19,23].

N was early termed an RNA chaperone[24] indicating its interaction with literally all types and sequences of RNA, and its potency of melting structured RNA in a functional context[25]. The modular character and high degree of disorder in N might favor promiscuous RNA-

[1]Institute for Molecular Biosciences, Goethe University Frankfurt, Max-von-Laue-Str. 9, 60438 Frankfurt/M., Germany. [2]Center for Biomolecular Magnetic Resonance (BMRZ), Goethe University Frankfurt, Max-von-Laue-Str. 7, 60438 Frankfurt/M., Germany. [3]European Molecular Biology Laboratory (EMBL) Hamburg Site, c/o Deutsches Elektronen-Synchrotron, Notkestr. 85, 22607 Hamburg, Germany. ✉e-mail: schlundt@bio.uni-frankfurt.de

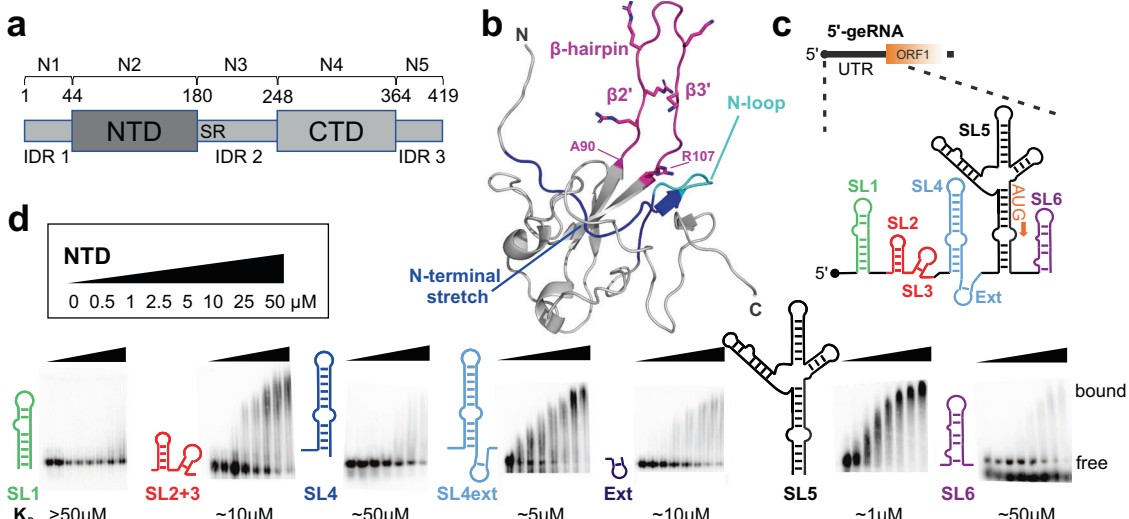

**Fig. 1 | The Nucleocapsid NTD of SCoV-2 exhibits differential interactions with the virus' 5'ge RNA elements. a** Domain organization of SCoV-2 N with numbers indicating domain boundaries. On top, an alternative domain nomenclature is given. **b** The N NTD (aa 44–180) as taken from PDB entry 6YI3[8]. Magenta highlights the β-hairpin, also termed basic finger, and sidechains of positively charged residues are shown with sticks. Labeled in blue is the N-terminal stretch (aa 47–66) and in cyan, the therein comprised N-loop finger (aa 58–63). **c** Scheme of the SCoV-2 5'ge RNA comprising the regulatory RNA elements SL1–6 (covering nt 1–343) including the 5'-UTR. The start codon (AUG) of ORF1 is shown in orange. **d** EMSAs showing the differential interactions of the NTD with the 5'ge RNAs SL1 to SL6. Protein concentrations of respective titration points are given in the inset. $K_D$ estimates are indicated below. See also Table 1, Supplementary Fig. 1 and Supplementary Table 1. Source data (all EMSAs with $n = 3$ technical replicates) are provided as a Source Data file.

binding[23,26–28]. Indisputably, full functionality requires a sensor for particular RNAs like packaging signals (PS), transcription-regulatory sequences (TRS) and other, still enigmatic regulatory elements. Literature extensively describes highly conserved, functional structures in the SCoV-2 genomic RNA (gRNA), specifically within the regulatory genomic ends[29–32], both in vitro and in virions/infected cells[33–36].

A recent study from 2020 revealed particular RNA sequences within the SCoV-2 5'-genomic end (5'ge) bound by N and claims those are targeted for packaging[17]. As shown in Fig. 1c, the highly conserved 5'-untranslated region (UTR)[37] as part of the 5'ge comprises a set of cis-regulatory RNA elements (5'ge RNA elements), termed 5' stem-loops (SL) 1 to 5, with the start codon located at the end of SL5. Additional 5'ge RNA elements follow starting from SL6 and beyond, in parts resolved on the secondary structure level[31] (Supplementary Fig. 1).

Surprisingly little is known about specific RNA-recognition by N. Studies have used intentionally short, random or bulk RNAs to derive determinants of preferences with very heterogeneous/contradictive outcome[8,11,19,38–42]. Thus, N target RNA selectivity has remained enigmatic seeing the large pool of RNAs in an infected cell, but likely involves multivalency as the basis for high affinity[26]. Much effort has been undertaken to establish that IDRs 1 and 2 increase affinity by non-specific interactions[26], rather than exhibit specificity. Hence, it is plausible to attribute specificity – needed to distinguish regulatory sequences like PS or TRS – to the folded NTD.

Several studies support this assumption by claiming the NTD to express specificity for the TRS in CoVs[12,43]. In a pioneering study, Dinesh et al. provide an atom-resolved model of TRS recognition by the NTD. They suggest the interaction as little-specific and preferential binding to rely on the ssRNA nature[8]. In line with this, a recently solved complex structure with a short, unrelated dsRNA reveals the NTD to interact exclusively – and thus non-specifically – with the phosphate backbone, giving no hint at drivers for specificity[42]. A study by Redzic et al. showed NTD binding to both ss and stem-looped RNA by NMR, indicating a preference for a GU-rich single strand, but taking into account the context of NTD-flanking IDRs[40]. Of note, TRS-derived ssRNA sequences in recent studies were examined outside their natural SL context[8,11,40]. Consequently, we lack a systematic comparative

analysis of the NTD's potential to differentially engage with the manifold types of regulatory viral RNA elements.

We here specifically focus on the role of the NTD in mediating target specificity, to approach the complex interactome of N with RNA. We provide insight into the differential interactions of N's NTD with the 5'ge RNA cis-regulatory elements SL1-SL6. Going beyond its promiscuity, we particularly concentrate on binding to the previously described regulatory sequences in SL3 and a region extending SL4 in 3' (Ext, Supplementary Fig. 1), both contrasting the weaker interactions with the 5'-UTR robust stem-loops SL1, 2, and 4. Importantly, Ext, together with additional downstream regions, was recently found to significantly crosslink to N[17], suggesting them to encode unique features for NTD specificity. We use NMR spectroscopy to identify a particular atom-resolved NTD signature provoked by the binding of those RNAs. In addition, we use analytical SEC, SAXS, and CD spectroscopy to manifest unique, stable complexes between NTD and SL3/Ext, while we show that both regions have transient RNA-structural propensities. We suggest that the comparable sequence compositions encode a unique element with respect to its structural dynamics, which is used by N's NTD for distinct recognition in the genome. By this, NTD exploits an additional level, given by complex stability that is key to selectivity within the broadly recognized landscape of RNAs.

## Results

### Differential preferences of SARS-CoV-2 N NTD for RNA elements
We initially examined the RNA-binding preferences of the NTD (aa 44–180) among individual SCoV-2 5'ge RNA elements in EMSAs (Fig. 1). We included the first 343 nt of the gRNA (Fig. 1c, Table 1 and Supplementary Table 1 and Fig. 1), comprising the 5'-UTR (nt 1–265, SL1–5) and the first 78 nt of the ORF1a/b coding sequence (SL6). We focused on individual elements to interrogate specificity-driving molecular parameters and, in this study, did not systematically examine the influence of various flanking contexts to all elements. The NTD differentiated between the subset of RNAs, judged by the broad range of apparent affinities (Fig. 1d). The low NTD concentration sufficient for complex formation with the large SL5 suggested a correlation of affinity with RNA size, and interactions to be driven by electrostatics.

**Table 1 | Overview of RNAs used in this study**

| RNA | Boundaries (genomic sequence position) | Number of nts | MW (kDa)[a] | Comment |
|---|---|---|---|---|
| *SARS-CoV-2 RNAs* | | | | |
| SL1 | 7–33 | 27 | 8.6 | - |
| SL2 + 3 | 40–80 | 43 | 13.7 | +2 additional 5′ G |
| SL4 | 86–125 | 44 | 14.1 | +2 additional GC bp[b] |
| SL4ext | 83–149 | 69 | 22.1 | +2 additional 5′ G |
| Ext | 129–148 | 22 | 7.1 | +2 additional 5′ G |
| Ext_C to A | 129–148 | 22 | 7.1 | Like Ext, with C140A |
| SL5 | 149–297 | 150 | 48.3 | +1 additional 5′ G |
| SL6 | 302–343 | 46 | 14.7 | +2 additional GC bp[b] |
| P2 | 726–756 | 33 | 10.7 | +2 additional 5′ G |
| SL1_ext[SL1] | 7–44 | 38 | 12.1 | - |
| SL1_ext[SL4] | 7–33 / 129–148 | 47 | 15.0 | - |
| SL4_ext[SL1] | 86–125 / 34–44 | 51 | 16.3 | - |
| P3 | 20668–20715 | 48 | 15.5 | - |
| P3_A | 20668–20692 | 25 | 8.1 | - |
| P3_B | 20691–20715 | 25 | 8.0 | - |
| *Additional RNAs* | | | | |
| ss19T | - | 19 | 6.1 | - |
| ss19B | - | 19 | 6.1 | - |
| ds19 | - | 19 (ds) | 12.3 | - |
| SL_AUA | - | 20 | 6.5 | - |

Sequences are listed in Supplementary Table 1.
[a]All theoretical MWs are calculated assuming 5′ monophosphate.
[b]+2 additional 5′ Gs and 2 additional 3′ Cs.

However, comparison of the NTD's preferences among smaller RNA elements revealed a significant grading. We found apparent micromolar affinities (~10 μM) for SL2 + 3 containing the TRS-sequence, involved in transcriptional regulation[8,12,25]. Although similar in size (Table 1), the affinity of NTD for SL4 was apparently lower (~50 μM). This demonstrates NTD-RNA interactions are not exclusively driven by net charge (thus scaling with RNA size), but include determinants for specificity. Strikingly, extending SL4 by the AU-rich stretch (Ext) located between SL4 and 5 (SL4ext) strongly increased the apparent affinity (~5 μM) compared to SL4 alone.

To obtain a more detailed understanding of the NTD interacting with the 5′-UTR elements, we used solution NMR on the basis of an NTD backbone assignment (Supplementary Fig. 2a). We expected the EMSA-observed affinity range to be reflected in corresponding $^1$H-$^{15}$N-chemical shift perturbation (CSP) plots. Unexpectedly, the NTD CSP pattern was similar for all 5′ge elements, and significantly shifting amides clustered with the positively charged surface (Supplementary Fig. 2b, c). Still, slightly higher average CSPs for SL2 + 3, SL4ext and Ext supported differences in affinity and, possibly, binding-mode. This was substantiated by the different line-broadening of NTD peaks with RNAs of the SL4 hub (i.e. SL4, SL4ext and Ext, Supplementary Fig. 3a). Preferentially bound SL4ext and Ext showed two significant intensity-dips locating to the N-terminal stretch (aa 47–66) and the β-hairpin. Additionally, the two RNAs are in intermediate exchange with the NTD, contrasting the fast exchange regime of SL4 (Supplementary Fig. 3b), representing distinct affinities. In support of that and in line with EMSAs, integrated dissociation constants ($K_D$) derived from ITC

experiments in both directions of titrating (Supplementary Fig. 4) were 53 μM for SL4, while 0.5 μM and 99 μM for SL4ext. For the latter we conclude the two binding events reflect the contributions from Ext and SL4, respectively. Altogether, these data unambiguously reflect the graded affinities of the NTD for 5′ge RNAs.

**Complex stabilities correlate with particular CSP signatures**

To gain insight into differential complex formation of the NTD with 5′ge RNA elements, we concentrated on the bifacial SL4 hub. We also included SL2 + 3 comprising the TRS-sequence based on its obvious role as an established NTD target. Conceiving the NTD as a hand-like fold, RNA-interacting regions locate to the central palm-like β-core and fingers, i.e. loops extending from it (Fig. 1b)[8]. We assumed these fingers are highly sensitive to RNA-induced changes. We picked residues located in the fingertip regions to serve as reporters for structural changes in the NTD (Fig. 2a). {$^1$H}$^{15}$N hetNOEs confirmed our reporter residues locate to regions of increased flexibility, most prominently within the β-hairpin (basic finger) and adjacent N-terminal loop (N-loop finger) (Fig. 2b). Similar to the homolog SCoV NTD[44], we assumed a correlated mobility of the N-loop and the basic finger (Supplementary Fig. 5).

We next used analytical size-exclusion chromatography (aSEC) to define complex stabilities of the NTD with SL2 + 3 and elements of the SL4 hub (Fig. 2c, Supplementary Fig. 6). NTD formed complexes with SL2 + 3, SL4ext and Ext as indicated by separate RNP elution peaks, unlike SL4 that did not form a stable complex with NTD under the experimental conditions. In accordance with results from EMSAs and CSPs, SL4ext seems preferred over Ext alone as indicated by different proportions of unbound RNA. This suggests a role of an RNA-structural context for NTD-binding.

We wondered if differential complex stabilities correlate with distinct NMR chemical shifts and in-depth-analyzed the reporter residues in HSQC-based titrations. We found a striking correlation of CSP trajectories with the type of complexed RNA (Fig. 2d, Supplementary Fig. 5b). For example, the basic finger-residues G99 and M101 display diametrically opposed trajectories with weakly bound SL4 compared to stable-complex-forming SL2 + 3, SL4ext and Ext, which share identical CSP trajectories. We concluded that different RNAs provoke distinct CSP signatures for them. A second group of peaks (e.g., G60, G96) displays RNA-individual CSPs. The shift changes of a third set of peaks (e.g., the terminal G44, G179 and S180) appear RNA-independent with the same trajectories but different maxima, in line with the approximate relative affinities.

The distinct RNA-dependent signatures are well confirmed by the plot of absolute $^1$H CSPs of basic finger residues 90 to 104 (Supplementary Fig. 5c), which reveal a co-occurrence of values and signs for SL2 + 3, Ext, and SL4ext compared to SL4 alone. The particular capacity for the basic finger as a sensor appears plausible taking into account the role of its intrinsic flexibility for RNA-binding[8].

We conclude that SL2 + 3 and the Ext RNA flanking the stable SL4 stem-loop comprise a particular sequence context for preferred interaction with NTD. Both elements are possibly recognized in a distinct manner as compared to the bulk of other RNA types and sequences in the 5′ge.

**Mutational analysis shows sensory capacity of reporter residues**

Next, we sought to show that the NTD reporter residues efficiently sense different RNAs rather than exhibit a direct involvement in RNA selection. We exchanged glycines in the basic (G99) and the N-loop (G60) fingers to bulky isoleucines (Fig. 3a). R107A – described as RNA-binding deficient mutant[8,45] – served as a loss-of-affinity control. Additionally, we replaced S105, located in the basic finger hinge region, with isoleucine, rationalized with the abolishment of a polar contact between the serine sidechain and the N-loop (Fig. 3a and Supplementary Fig. 7). We expected the mutation to impair the correlated

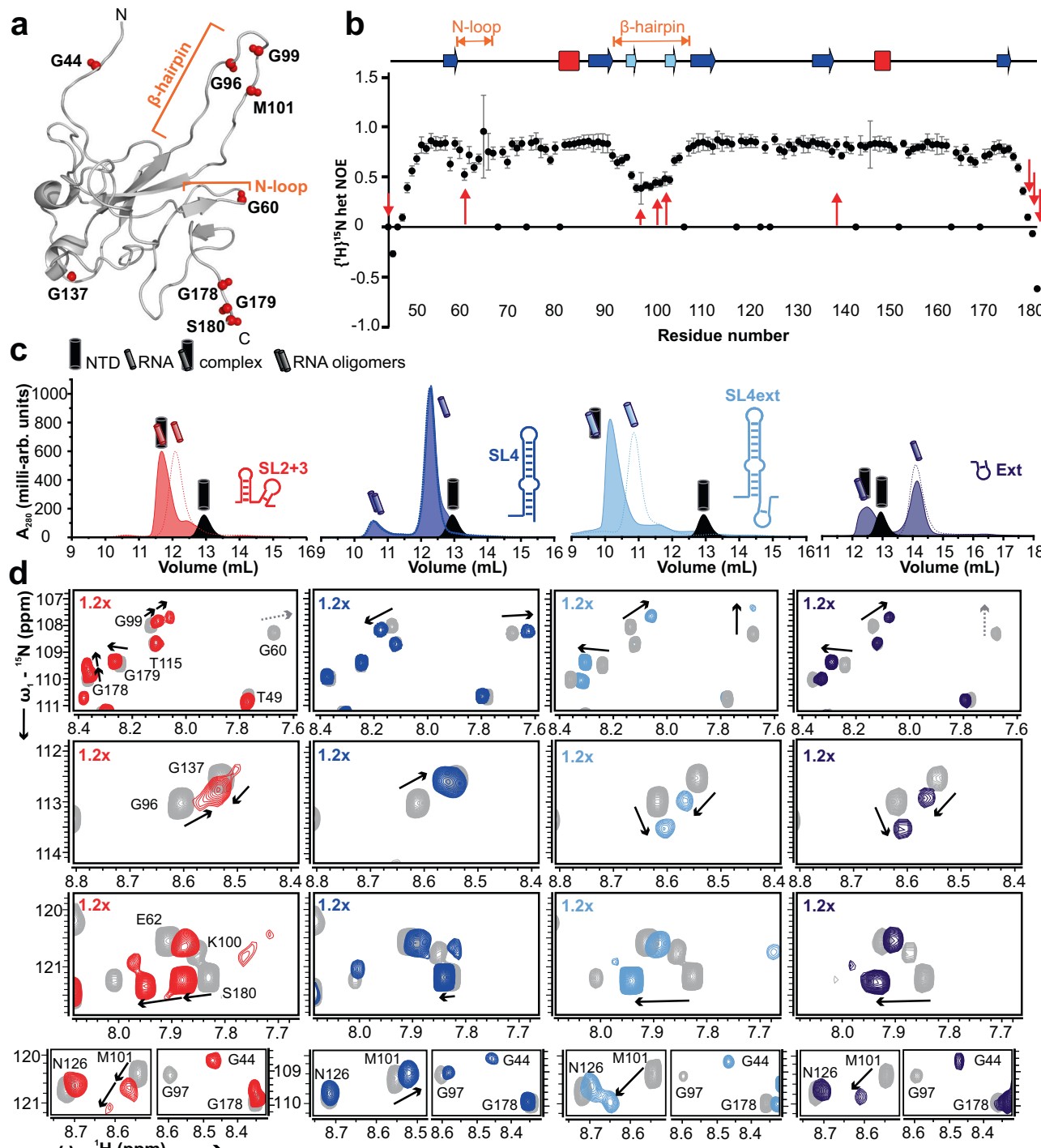

**Fig. 2 | Preferred complexes of the NTD with 5'ge RNA elements are sensed by NTD reporter residues. a** Selected reporter residues of RNA-sensing NTD fingertips are shown as red spheres (N-H groups). The N-loop finger (N-loop) and basic finger (β-hairpin) are labeled for orientation. **b** {¹H}¹⁵N hetNOE ratios as measured at 298 K display increased mobility of distinct NTD regions ( < 0.6). Errors are derived for a single experiment from the program CCPNMR Analysis 2.4[76] based on the respective signal-to-noise of spectra. Reporter residues shown in (**a**) are indicated by red arrows. Secondary structure elements are depicted above. See also Supplementary Fig. 5a. **c** aSEC runs at RT of NTD with 1.2-fold molar excess of 5'ge RNA elements (filled curves) showing SL2 + 3 (red), SL4 (blue), SL4ext (light blue) and Ext (dark blue), respectively (A₂₈₀ absorbance plotted over retention volume). Corresponding runs of NTD alone (black, filled curves) and RNA only (respective color, dotted lines, open curve) are included. See also Supplementary Fig. 6. **d** ¹H-¹⁵N-HSQC zoom-ins of NTD showing reporter residues from panel a and their CSP signatures (indicated by arrows) upon titration with 1.2x molar ratios of SL2 + 3 (red), SL4 (blue), SL4ext (light blue) or Ext (dark blue) at 298 K. See also Supplementary Figs. 2, 3, and 5. Source data are provided as a Source Data file.

movement of the two fingers and affect RNA discrimination by the NTD.

We compared the ability of NTD wildtype and NTD mutants to distinguish between SL4ext and the less preferred SL4 by means of CSP

signatures (Fig. 3b–e). Indeed, the S105I mutant shows a clearly impaired ability to discriminate between SL4 and SL4ext (Fig. 3c), indicating that communication between fingers is essential for functionality. Neutralization of the central palm residue R107 to alanine

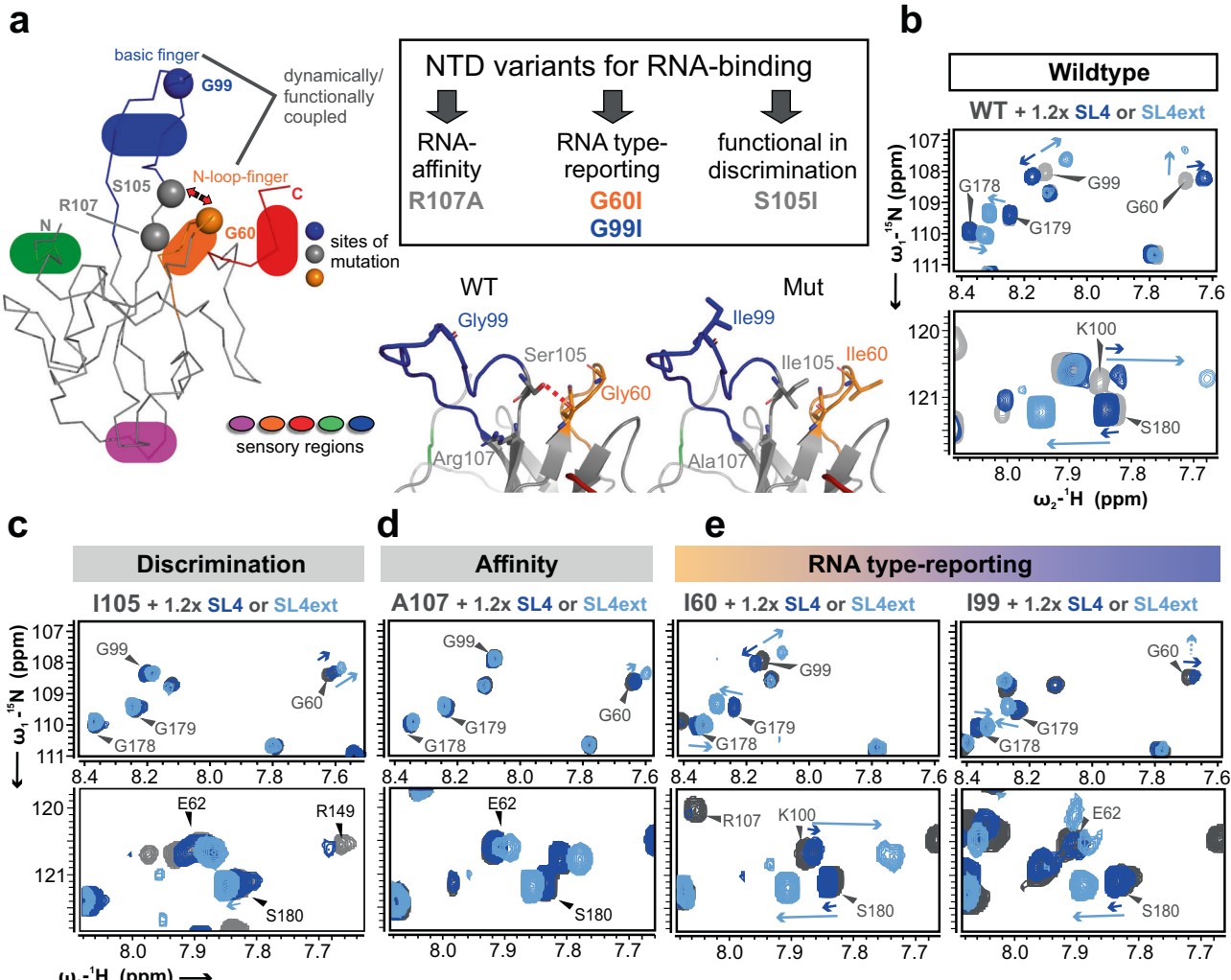

**Fig. 3 | The sensory capacity of specific NTD residues for preferred RNA complex formation is confirmed by mutations. a** Mutants were selected based on the NTD hand-like structure, with a focus on flexible fingers and categorized into three classes: affinity, type-reporting, functional discrimination. The zoom-in of the structure suggests a functional contact between the side-chain of the basic finger hinge residue S105 and N-loop residue Q58 in the WT NTD, while abolished in the S105I mutant. The pictures were made from PDB entry 6YI3[8] and WT-residues replaced by the given ones in Pymol 2.0 (Schrödinger). **b–e** $^1$H-$^{15}$N-HSQC zoom-ins for the three categories of NTD mutants as introduced in panel a showing protein alone overlaid with spectra from NTD complexes with preferably bound SL4ext or non-specifically bound SL4. CSP signatures of labeled reporter peaks as introduced in Fig. 2 were used as read-out for: the WT control (**b**), the functional/discrimination-impaired mutant S105I (**c**), the loss-of-affinity mutant R107A (**d**), and the two reporter residue mutants G60I and G99I (**e**). All data are recorded at 298 K. See also Supplementary Fig. 7. Source data are provided as a Source Data file.

effectively abolished interaction with SL4 (Fig. 3d). Interestingly, although significantly reduced, R107A was still able to bind SL4ext. This supports the idea that specificity in recognition of SL4ext is provided beyond pure electrostatics. In contrast to these functional mutants, reporter mutants G60I and G99I kept their ability to discriminate between SL4ext and SL4, shown by distinct CSP signatures of the remaining reporter residues (Fig. 3e). Altogether, these data support the applicability of reporters to predict preferred RNA targets in a simple HSQC-based experimental setup. Further, the discrimination-deficient S105I mutant reveals an important role in dynamics within and between individual NTD regions. The proposed intramolecular communication between the fingers is in line with significant effects on CSPs of these regions in S105I spectra compared to WT (Supplementary Fig. 7).

**SAXS confirms selective complex formation with viral 5'ge RNA**
To describe complex formation of NTD with SCoV-2 5'ge RNAs directly on the RNP-level, we applied small-angle X-ray scattering (SAXS), a solution method combinable with SEC that yields low-resolution

structural information, thus complementing NMR. SEC-SAXS allows analyzing individual fractions of NTD, RNA, and RNPs, and their contributions in mixed states. The quality of the SAXS data is seen from excellent fits of experimental scattering for free NTD and RNAs with calculated curves from the atomic coordinates of an NTD monomer[41] or monomeric RNA models, respectively (Supplementary Fig. 8a–d). Consequently, all SEC-SAXS-derived structural parameters for NTD and RNAs are consistent with the corresponding parameters obtained from atomistic models of the individual components (Table 2).

We especially ensured the monomeric NTD state[8,40,46], independent of concentration, by SEC-MALS and aSEC in our physiological/NMR (phosphate) buffer system (Supplementary Fig. 8b, c). We further confirmed the monomeric state of the NTD and RNA components by SEC-SAXS in a HEPES buffer system optimized for SAXS, additionally using SEC-MALS (see Supplementary Methods and Supplementary Fig. 8f). This was done to rule out non-specific effects from radiation damage in phosphate buffer[47,48]. Importantly, SAXS data acquired in HEPES show near-complete overlay with measurements in phosphate for all components, underlining the integrability

**Table 2 | Summary of SEC-SAXS data recorded on the free and complex samples as shown; frames are provided in Supplementary Fig. 8**

| Protein | RNA | MW (kDa) Theor. Exp. | $D_{max}$ (nm) Theor.[b] Exp. | $V_P$ (nm³) Exp. | $R_g$ (nm) [a] Theor.[b] Exp. |
|---|---|---|---|---|---|
| NTD | | 14.9   13.5 | 5.7   6.1 | 23.4 | 1.6   1.6 |
| | SL2 + 3 | 13.7   14.2 | 8.1   8.0 | 24.1 | 2.3   2.2 |
| | SL4 | 14.1   14.8 | 7.5   7.1 | 21.6 | 2.0   2.1 |
| | SL4ext | 22.1   27.6 | 10.0   10.2 | 32.9 | 2.8   2.9 |
| | Ext | 7.1   8.5 | 4.6   5.4 | 12.6 | 1.3   1.6 |
| NTD | SL2 + 3 | 28.6   23.1 | n.a.   9.0 | 37.5 | n.a.   2.5 |
| NTD | SL4 | 29.0   18.1 | n.a.   7.8 | 24.9 | n.a.   2.2 |
| NTD | SL4ext | 37.0   36.1 | n.a.   13.5 | 50.6 | n.a.   3.6 |
| NTD | Ext | 22.0   17.4 | n.a.   8.5 | 32.3 | n.a.   2.2 |

n.a. No suitable RNP models utilizable. [a]The given $R_g$ values were obtained from the $P(r)$ analysis. [b]Theoretical $D_{max}$ and $R_g$ as obtained from the Crysol-derived envelope values. All numbers are given with one decimal place only for convenience here.

A comparison with theoretical values is given for free components. SAXS data quality estimates and details of measurements are given in Supplementary Table 2.

of SAXS with all other experiments of this study in the physiological buffer system.

To probe complex formation in conditions consistent with our NMR analyses, we measured SEC-SAXS data from RNA-contained fractions in standard phosphate buffer for SL2 + 3, SL4, Ext, and SL4ext RNAs alone and together with NTD (Supplementary Fig. 8g) and monitored changes in the radius of gyration, $R_g$, scattering pair distance distributions, $P(r)$, as well as overall volumes and MWs caused by NTD/RNA association (Fig. 4, Table 2, Supplementary Table 2). Relative to the $R_g$ of the individual RNA components our analysis reveals substantial increases in the $R_g$ – of almost 0.7 nm – and corresponding molecular weights for the NTD-associated Ext and SL4ext RNP complexes (Fig. 4a, b), as well as changes in the $P(r)$ profiles (Fig. 4c), indicating the formation of complexes of NTD with these preferred RNAs. In contrast, it appears that for SL4 the scattering of both samples is exclusively caused by the RNA component, where both the SL4 and the NTD-SL4 SAXS profiles appear near-identical and generate very similar structural parameters and $P(r)$. This underlines the sole presence of non-bound RNA in the respective fractions even in the presence of NTD (Supplementary Fig. 8g) and is best seen by the lack of increase in MW in the presence of NTD, different from the other three RNAs (Fig. 4a). Interestingly, for the NTD-SL2 + 3 complex an increase in the apparent molecular weight is observed, indicative of RNP formation, but this is accompanied by only a moderate increase in $R_g$ (~0.2 nm) compared to the Ext and SL4ext RNPs (Fig. 4). This may suggest differences in the modes of NTD/RNA interactions, e.g., conformational rearrangements of the RNA on complex formation, or different positioning of the NTD relative to the center of mass of the corresponding RNA within the complex. The differential complex formation is also supported by the change in $P(r)$ profiles and subsequently $D_{max}$ when comparing Ext and SL4ext to SL2 + 3. Here, the latter clearly reflects the possible rearrangement compensated by the NTD-binding as expressed by similar profiles and maximum distances, while both Ext and SL4ext reveal significant increases in complex dimensions. In line with the $R_g$ analysis, SL4 profiles show no differences for SEC-SAXS frames between RNA alone and in the presence of NTD, while SL2 + 3, Ext, and SL4ext show clear shifts of RNA-contained frames (Supplementary Fig. 8f). Altogether, the SAXS confirms the potential for NTD to differentiate between natural 5'ge RNA elements as measured by changes in the global structural parameters of the resultant RNP complexes in the comparative NMR sample conditions.

## Motif-specific vs. electrostatic interactions of NTD-RNA

Our data suggest the NTD interacts with particular viral RNA elements containing a specificity determinant that may also be influenced by, and in combination with, more-generalized electrostatic interactions that often drive RNA/protein binding. To probe the differential impact of charge for NTD engaging with RNAs of the SL4 hub, we quantified RNA-binding as a function of salt concentration. A similar approach was applied recently for specific HIV-1 genome packaging[49]. We used NMR to follow salt-dependent CSPs of reporter residues and total spectral intensities of individual RNPs as read-out (Fig. 5a–c, Supplementary Fig. 9). Complexes were formed at low salt (50 mM KCl) to foster RNP formation, and significant CSPs of reporter residues were found for all three RNAs (Fig. 5a). Salt-resistance of RNPs was subsequently monitored by increase of [KCl] to a final concentration of 405 mM. Differential reduction of RNA-binding at high-salt was found for the three RNAs, reflected by CSPs for residues located either in the basic (T91, G99, M101), the amino-terminal or N-loop fingers (D47, G60, V72) (Fig. 5b). The NTD complex formed with SL4 was most vulnerable to salt, indicated by the steepest decline of CSPs. No SL4-induced CSPs remained visible at 405 mM KCl, while Ext and SL4ext still caused up to 20% of the maximum CSP (Fig. 5b). Interestingly, salt-dependence is most pronounced for residues in the basic finger (T91, G99, M101). This is supported by RNA-induced line-broadening (Fig. 5c). A significant intensity loss remains visible for N-loop and basic finger with SL4ext and Ext, comparable to the pattern at 150 mM KCl (Fig. 5c, Supplementary Fig. 3). In contrast, the average peak intensity in presence of SL4 is almost identical to free NTD, indicating complete disassembly of the complex. Finally, the influence of salt on differential complex stability was confirmed by ITC. As seen in Supplementary Fig. 4b, an increase in KCl concentration leads to a loss of binding when titrated to SL4, which was weak but detectable (64 μM) at low-salt conditions (Supplementary Fig. 4a). In contrast, SL4ext still shows binding to NTD at high-salt conditions, where likely the Ext part of SL4ext is still recognized, albeit with lower affinity. This observation is well in line with the affected linewidth caused by a shift in the exchange regime from intermediate to fast observed for SL4ext by NMR at high-salt conditions (see Source Data). We conclude that the NTD interaction with SL4 is primarily driven by charge, while the interaction with SL4ext and Ext includes additional determinants of specificity and affinity.

## Complex formation with RNA alters NTD dynamics

Recently, the NTD was found to comprise regions with high-amplitude dynamics on a ps-ns timescale (e.g. the basic finger)[40,44]. Interestingly, those dynamics were not significantly affected by a high excess of a DNA 7mer, likely owing to moderate affinity. To examine effects induced by a more affine-bound viral target we compared NMR-derived backbone dynamics of the NTD alone and in complex with Ext RNA on multiple time scales: via measurements of the steady-state hetNOE and of $^{15}$N relaxation rates $R_1$ and $R_2$ (Fig. 5d). RNA-mediated effects on NTD dynamics cluster with the two main RNA-interacting regions, i.e. the N-loop finger - and the basic finger. While N-loop residues mainly gain flexibility, suggested by reduced NOE values, the opposite is seen for basic finger residues. In complex with Ext, basic finger residues display on average 23% increase in NOE values, expressing a significant loss of motion (Fig. 5d). This is in accordance with previously suggested correlated motions of both fingers[44], which here are disrupted upon RNA-binding.

The underlying ns-range is ideally probed in more detail by $^{15}$N longitudinal ($R_1$) and transversal ($R_2$) relaxation rates of NTD alone and in complex with Ext. We found that NTD $R_1$ values for flexible regions, such as the N-loop and the basic finger, are up to 26% elevated above the average of 1.5 s$^{-1}$. This effect is largely obliterated in complex with Ext, where basic finger residues share the domain average $R_1$ of 1 s$^{-1}$ or lower (Fig. 5d). Notably, we found the same tendency within the N-loop

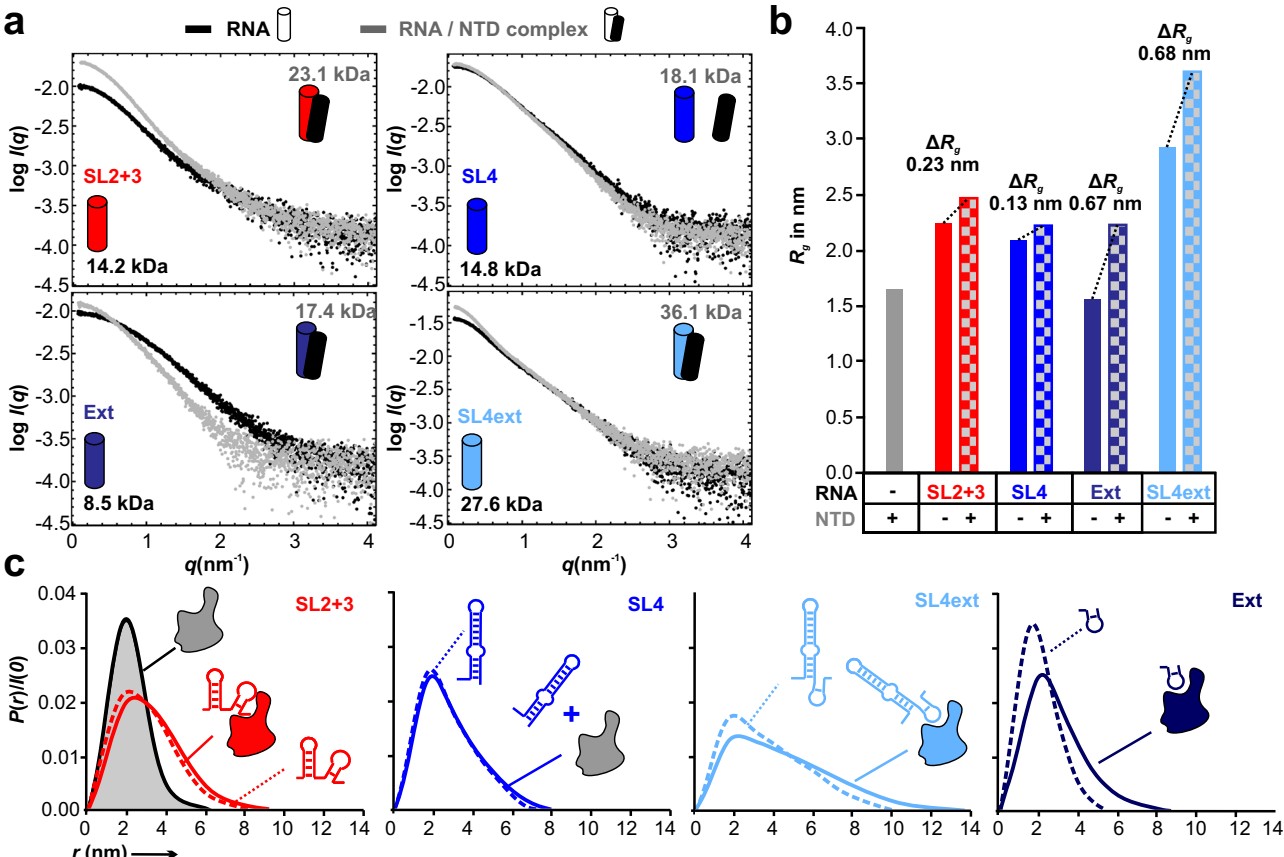

**Fig. 4 | SAXS confirms differential NTD complex formation with SCoV-2 5'ge RNAs on the level of RNP shapes. a** SAXS data comparisons of the free RNA components (black) and their corresponding NTD-complexes (RNPs, grey). Apparent molecular weights (rounded numbers, derived from Table 2) estimated for the SAXS data are given for RNA alone (black) and the respective RNPs (grey). The experimental MW for the NTD alone is 14 kDa (see Table 2 and Supplementary Fig. 8). **b** Plot of $R_g$ as derived from panel a for the NTD, SL2 + 3, SL4, Ext, and SL4ext, and changes in $R_g$ of the SL2 + 3, SL4, Ext and SL4ext RNP complexes, relative to the $R_g$ of the respective individual RNA components. Values were obtained in accordance with the $P(r)$ analysis in c. **c** Scattering-pair distance distributions ($P(r)$ profiles) as derived from panel a, calculated for the NTD, SL2 + 3, SL4, Ext, and SL4ext components and respective RNP complexes. The presence of complexes or mere mixes of RNA and NTD, as is the case for SL4, are indicated by icons. All shown data were recorded at 293 K in the standard NMR/phosphate buffer conditions. See also Table 2, Supplementary Table 2 and Supplementary Fig. 8. Source data are provided as a Source Data file.

finger, but observed a larger degree of deviations between single residues. For $R_2$, attenuating effects of the Ext RNA are less pronounced, possibly caused by incomplete saturation of the protein leading to additional exchange contributions in rate determination. Nonetheless, $R_1$ and $R_2$ values allowed calculating the correlation time $\tau_C$ for both species (Fig. 5e). A mean $\tau_C$ of 9.1 ns for the NTD results in an estimated molecular weight (MW) of 14.9 kDa, which perfectly fits the expected MW of monomeric NTD (see also Supplementary Fig. 8b, c). In complex with Ext, a mean $\tau_C$ of 13.8 ns correlates to 22.6 kDa, which is in good agreement with the expected MW of 22.0 kDa for a 1:1 complex. Our results indicate that complex formation of the NTD with Ext alters its dynamics, in contrast to non-specifically formed complexes with unrelated nucleic acids[40].

### Characteristic features of the SL2 + 3 and SL4ext RNA hubs

The two NTD-preferred 5'ge RNA elements SL2 + 3 and SL4ext include stretches with high A/U content (Fig. 6 and Supplementary Fig. 1). Interestingly, both SL3 and Ext had been suggested to exist in transient stem-looped conformations[31,33,34,50]. Furthermore, both are preceded by stem-loops conserved among coronaviruses, namely SL2 and SL4. We hypothesized that SL3 and Ext represent specific targets of NTD based on their intrinsic dynamics. We determined melting temperatures ($T_M$) of SL2 + 3 and the SL4 hub RNAs using CD-spectroscopy (Fig. 6a and Supplementary Fig. 10). For SL2 + 3 and SL4ext we

obtained two clearly distinct $T_M$ values, verifying them as tandems of a stable 5' moiety ($T_M$ values above 60 °C) and a labile 3' element ($T_M$ values below 30 °C). Notably, $T_M$ values for isolated SL4 (67 °C) and Ext (38 °C) RNAs were higher compared to those in SL4ext, likely due to construct design for in vitro transcription (Supplementary Table 1, Supplementary Fig. 1b). However, also steric effects contribute to mutual destabilization of both elements in the SL4ext context (Supplementary Fig. 11a, b). To probe for RNA secondary structure, we next recorded ¹H-1D NMR spectra (Fig. 6b and Supplementary Fig. 11). Imino proton resonances unambiguously report base-pairs assignable to nucleotides in stems. While the lack of imino signals does not necessarily represent full absence of RNA structure, it indicates significant portions to be single-stranded or in exchange. Indeed, the lack of imino peaks for the SL3 and Ext moieties at 298 K supports the labile stem-loop nature of both RNAs suggested by CD. For both, imino peaks are visible below ~290 K (Fig. 6b and Supplementary Fig. 11) confirming their secondary structure propensity and an equilibrium of opened and stem-looped conformers at 298 K. In line with this, SL2 + 3 revealed a second conformation related to the folding of SL3 at lower temperatures (Supplementary Fig. 11). For Ext, we unambiguously assigned imino signals both in the isolated and in the SL4ext context as depicted for secondary structure predictions (Fig. 6c).

Complementing NMR, we expected SAXS to reveal those RNA dynamics of SL2 + 3 and the SL4 hub elements and at the same time

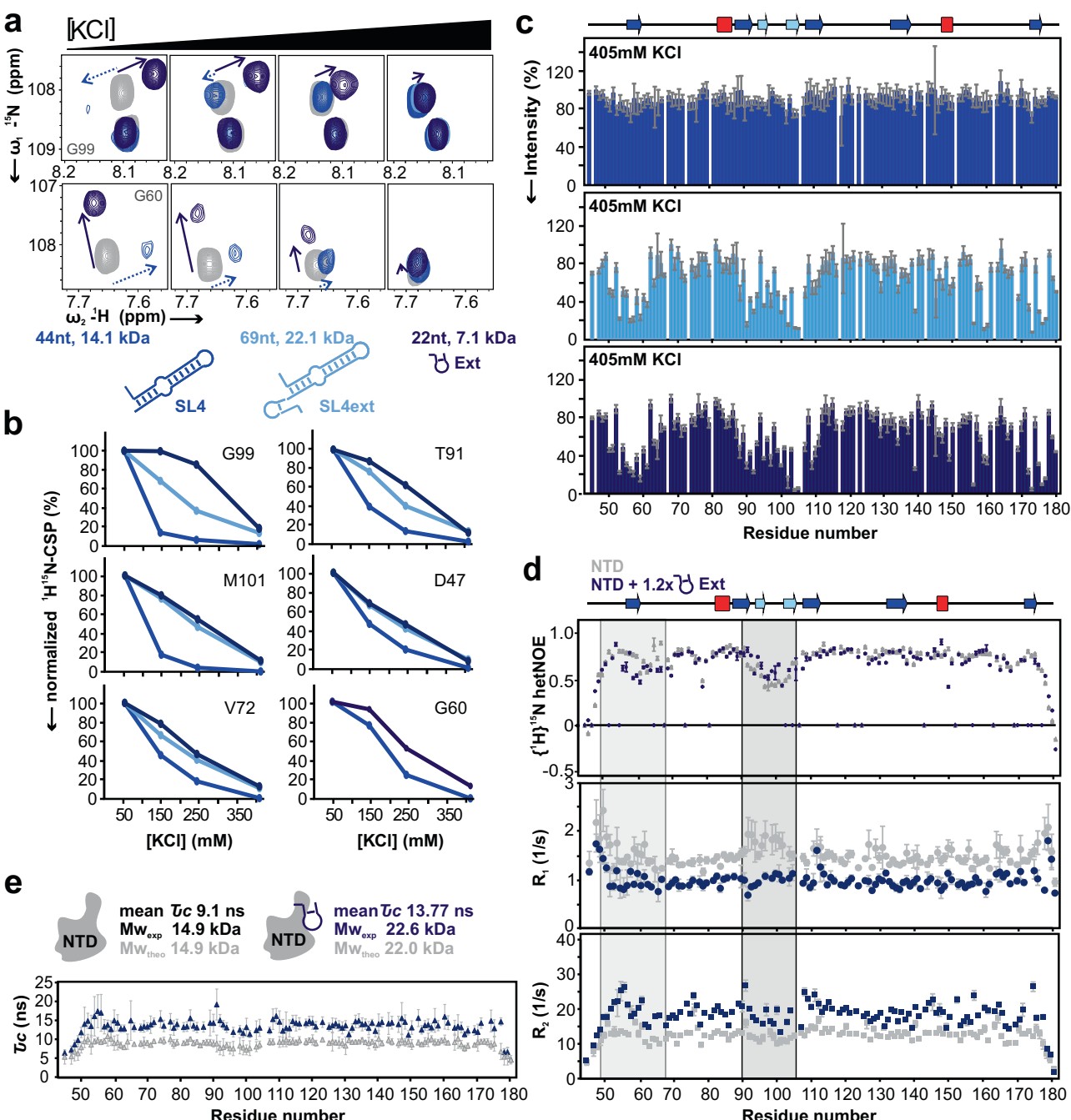

**Fig. 5 | Different contributions from charge-based interactions on NTD complex stability with the SCoV-2 SL4 hub RNAs. a** Two spectral excerpts of $^1H$-$^{15}N$-HSQC spectra overlaying free NTD (grey) with NTD in presence of 1.2-fold molar excess of SL4 (blue), SL4ext (light blue) and Ext (dark blue). Accordingly, panels indicate increasing KCl concentrations (from left to right: 50 mM, 140 mM, 240 mM and 405 mM KCl). The RNAs are shown as icons below with their sizes and molar weights. **b** Selected residues showing a salt-dependent decrease of combined $^1H$-$^{15}N$-CSP, normalized to respective maximal CSP values. **c** Normalized HSQC signal intensities at 405 mM KCl for NTD in complex with SL4 (blue), SL4ext (light blue) and Ext (dark blue). Values are derived from a single experiment and errors calculated using CCPNMR Analysis[76] reflecting the local signal-to-noise influence. See also Supplementary Fig. 9. **d** Dynamics of NTD (grey) versus NTD-Ext complex

(dark blue) expressed by $\{^1H\}^{15}N$ hetNOE and $^{15}N$ relaxation rates $R_1$ and $R_2$ obtained from single experiments, respectively. The highlighted areas indicate the N-terminal stretch (including the N-loop) and basic finger, respectively. For errors in the top panel, see panel c; errors of $R_1$ and $R_2$ represent the uncertainty of relaxation rate fits using CCPNMR Analysis[76]. **e** Rotational correlation times $\tau_C$ for NTD and the NTD-Ext complex plotted for individual residues. Errors are derived from propagation of errors in panel d, thus representing the mathematical uncertainty of ratios based on single datasets, respectively. Mean values and the derivable molecular weight, compared to the theoretical complex molecular weight, are given above. All data were recorded at 298 K. Source data (full spectra for salt titrations) are provided as a Source Data file.

report on their folds more globally. We filtered secondary structure-based models of the RNAs against SAXS data as a restraint for the global geometry and base pairing with RNAmasonry[51]. For SL2 + 3, we found the stem-loops in an atomic model would only fulfill

experimental SAXS data in a non-coupled arrangement (Fig. 6c), while co-axial stacking of the two parts has been suggested recently[52]. Possibly, this represents the particularly dynamic nature of the SL2 + 3 region, likely as a regulatory tool and potential hallmark for

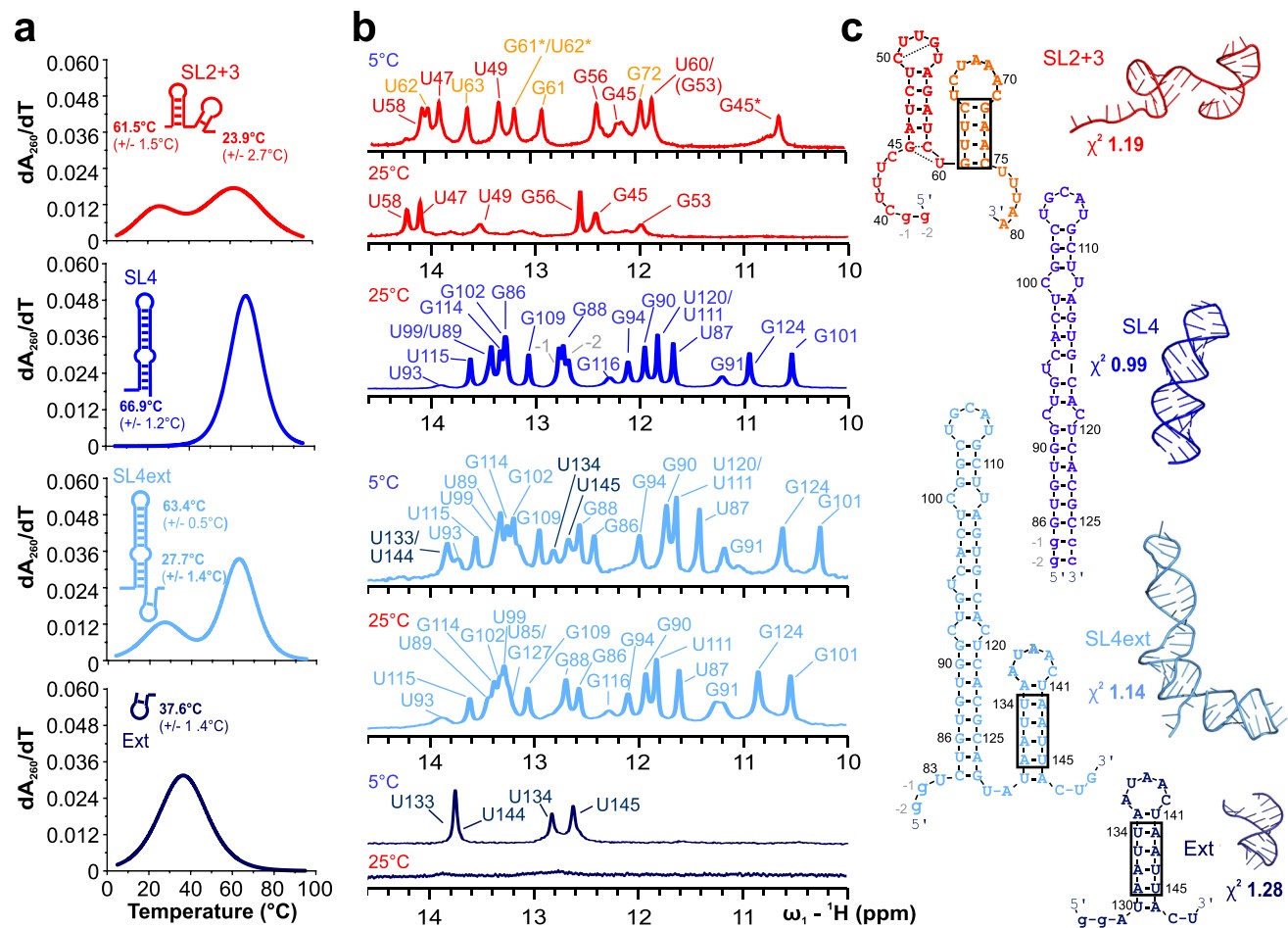

**Fig. 6 | Biophysical analysis of the SL2 + 3 and SL4 RNA hubs reveals transient folds in NTD-preferred RNA motifs. a** Representative melting curves of RNAs as labeled, plotted as the first derivative of 260-nm absorbance using CD spectroscopy. Peak maxima and melting points with errors are given with RNA icons, respectively (see also Supplementary Fig. 10). Source data (exact details for all RNAs with n ≥ 3 biologically independent as well as technical replicates) are provided as a Source Data file. **b** Imino-¹H-1D NMR spectra of SL2 + 3, SL4, SL4ext, and Ext with imino proton assignments, respectively, recorded at 298 K (25 °C) and 278 K (5 °C), the latter to visualize labile elements – boxed stems in panel c - that

lack imino peaks above 288 K. Orange labels (SL2 + 3) and dark-blue (SL4ext and Ext) labels highlight iminos derived from labile elements, SL3 and Ext, respectively. Asterisks in SL2 + 3 indicate a second conformation. See also Supplementary Fig. 11 for details and comments on RNA assignments. **c** RNA secondary structure predictions by RNAfold[89] as confirmed by NMR, and models derived from RNAmasonry[51] and supported by SAXS data for SL2 + 3, SL4, SL4ext and Ext, respectively. The fits of theoretical (as from the model) versus experimental SAXS data are given. See also Supplementary Fig. 8. Small letters indicate artificial nucleotides.

preferred targeting of SL3 by the NTD. As expected, we found SL4 to be a stable, enlarged stem-loop satisfying the suggested geometry (Table 2, Fig. 6c)[31]. We also found Ext to comprise a visible degree of structure, in good agreement with the model and the suggested equilibrium based on its T_M. The combined RNA SL4ext has been suggested to provide co-axial stacking of the two RNA subunits earlier[29]. We, however, found structural independence within the model based on SAXS at 293 K, supported by the clearly distinct melting points of SL4 and Ext within SL4ext in Fig. 6a. Altogether, combinatorial information obtained from SAXS, CD, and NMR indicate an independent assembly of SL4 and Ext in their tandem context with respect to a shared fold. For both SL2 + 3 and SL4ext our approach underlines the strong complementarity of NMR and SAXS for molecules in solution[53].

**NTD preference is not simply ssRNA over dsRNA**
Our findings on preference for the labile SL3 and Ext elements prompted us to ask whether RNA-binding of the NTD is merely a matter of discrimination between ss and dsRNA. Guided by the assumption that the NTD prefers binding to AU-rich sequences (see above and ref. 25), we chose a set of RNAs to examine the general role of ss vs. ds and stem-looped RNA. We used two 19mer RNA oligos

(ss19T and ss19B) with a 7mer A/U core motif that can be hybridized into a duplex RNA (ds19). We also included SL_AUA with a stable stem of 7 GC base pairs capped by an AU-rich hexaloop (Fig. 7a and Supplementary Fig. 1c and 12a). We carried out protein-observed NMR experiments and inspected CSP signatures for selected reporter residues as before (Fig. 7a). For convenient comparison, we schematically included peak positions for the weak binder SL4 and the preferred SL4ext in zoom-ins of individual titrations. Surprisingly, none of the tested RNAs – independent of their ss/ds/SL status – categorized as 'preferred', indicated by the SL4-like signatures of reporter peaks. This was further corroborated by their weak binding in EMSAs (Source Data) and instable complexes in aSEC as shown for ss19T (Supplementary Fig. 6).

In contrast, we tested another RNA that we expected to categorize as preferred binder. The P2 RNA element[17] is located in the ORF1a/b region between SL12 and 13 (Supplementary Fig. 1 and 12b). Consistently, both the Ext region between SL4/SL5 and P2 had been described as hotspots for cross-linking with N and are expected to specifically trigger LLPS in the context of genome packaging[17]. As shown in Supplementary Fig. 12c, P2-binding to NTD in the SL4ext-mode unambiguously supports the inherent robustness and validity of our combinatory approach. On the basis of our data we further

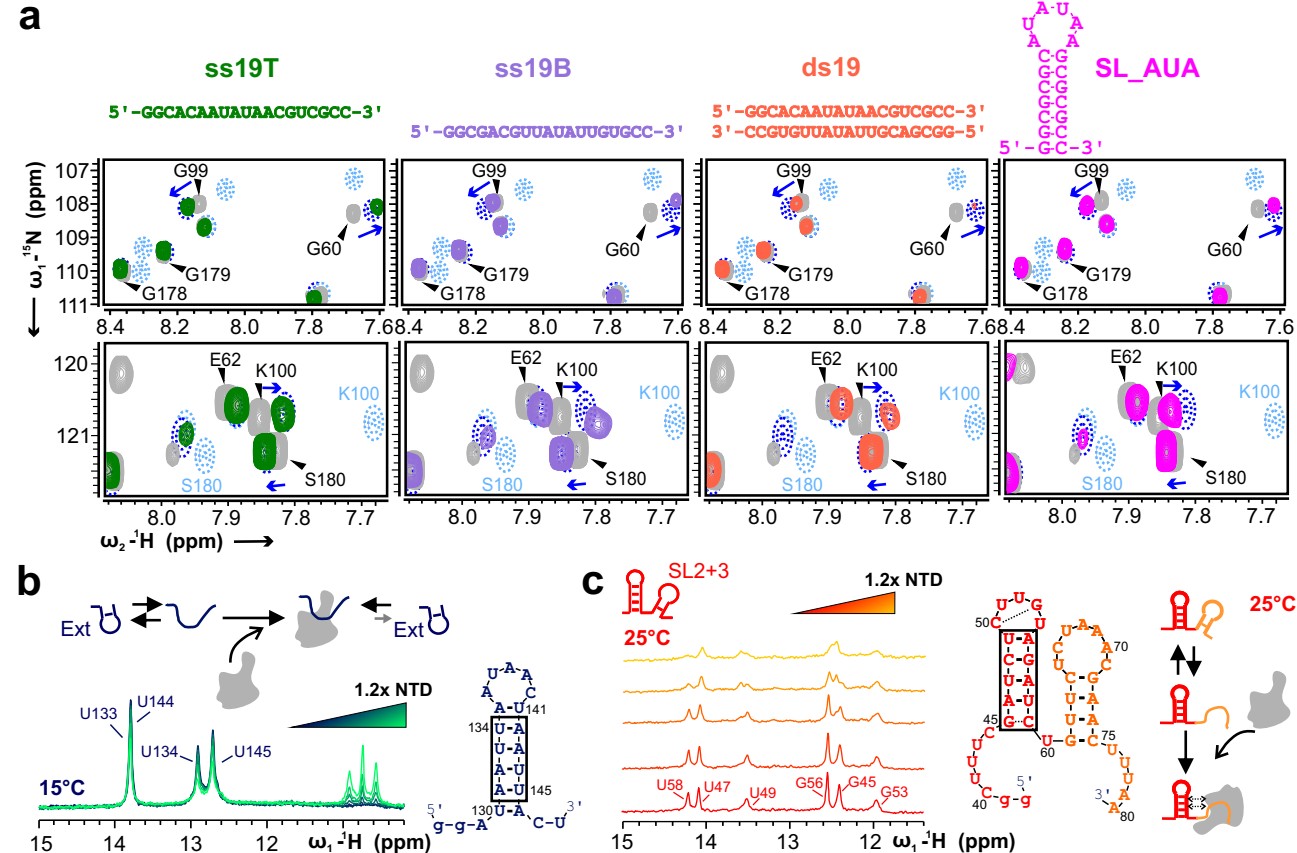

**Fig. 7 | Role of RNA secondary structure and AU-rich sequences for binding by NTD. a** Two spectral excerpts of ¹H-¹⁵N-HSQC spectra recorded at 298 K overlaying free NTD (grey) with various RNAs. The upper panel includes G99 and G60; the lower one the C-terminal S180. From left to right: 1.2x molar excess of ss19T (green), ds19 (orange), ss19B (purple) and SL_AUA (pink). For comparison of CSP signatures, chemical shifts of 1.2x SL4 (dotted blue lines) and SL4ext (dotted light blue lines) are schematized, respectively. **b** Imino-¹H-1D NMR spectra visualizing the folded conformer of Ext when titrated with increasing amounts of NTD at 288 K (15 °C). A scheme for the underlying mechanism is shown on top. Signals upfield of 11 ppm are protein-derived. **c** Imino-¹H-1D NMR spectra visualizing the folded SL2 + 3 RNA at 298 K (25 °C) when titrated with increasing amounts of NTD. The schemes show the genomic numbering of SL2 + 3 placed with its suggested secondary structure and the proposed mechanism of specifically engaging with SL3. In panels b and c, boxes indicate base pairs that are visible under the given experimental conditions. Small letters indicate artificial nucleotides. See also Supplementary Figs. 11 and 12.

investigated a SCoV-2 packaging signal described recently[54]. Within the 1-kb region located at the end of ORF1b, we found a 48nt SL4ext-like tandem RNA element, we termed P3 (Supplementary Fig. 1b and 13a). The stable 5′ end stem-loop, P3_A displayed SL4-like signatures of reporter peaks, while the labile P3_B and the tandem P3 could be classified as SL4ext-like binders (Supplementary Fig. 13b). Correlating complex stabilities as indicated by CSP signatures of reporters were confirmed by aSEC (Supplementary Fig. 13c).

We next asked whether any viral stem-loop element, extended by a subsequent single-stranded sequence was sufficient for specific recognition by the NTD. We thus tested SL1, followed by the naturally extending 11 nt (SL1_ext^SL1) (Supplementary Fig. 1d and 14a). We unambiguously categorized SL1_ext^SL1 as a non-preferred, SL4-like binder, indicated by reporter residue CSP signatures (Supplementary Fig. 14b). Identical to SL4, ITC confirmed the naturally extended SL1 as non-preferred binder with an affinity of only 87 μM at low-salt conditions and complete loss of binding at high salt (Supplementary Fig. 4). Interestingly, swapping the ext regions of SL1 and SL4 (SL1_ext^SL4 and SL4_ext^SL1) clearly revealed the extension as the key element for NTD recognition (Supplementary Fig. 1d and 14). The artificial SL1_ext^SL4 construct classified as SL4ext-like, preferred binder – in contrast to SL4_ext^SL1 – suggesting the particular characteristic of a labile element plays a larger role than the exact genome position. In sum, our results support the NTD's preference for ssRNA, over stably folded RNA elements. Yet we show that, similar to folded elements, non-relevant

ssRNA motifs, despite an AU-rich content, are bound in a non-specific mode.

### NTD binding persistently shapes equilibria of RNA foldamers
Next, preferential binding of unfolded viral elements was confirmed by NMR. We monitored Ext RNA imino peaks at 288 K during titration with NTD and observed no peak shifts or line-broadening (Fig. 7b). In turn, a stepwise decrease of imino signals suggested the NTD interferes with the equilibrium of Ext conformations, constantly capturing the unfolded, thereby diminishing the folded population. We asked, if the NTD would specifically affect the foldamer ratio or if the complex was subject to mere temperature equilibrium (Supplementary Fig. 15a, b). We compared imino signal intensities of a complex mixed at 278 K and after warm-up to 310 K and re-cooling to 278 K. We found a visible loss of signal after cycling, while the absolute intensity in the sample remained unaltered. This indicates that the NTD had engaged with the more accessible ssExt at 310 K and this complex remained stable after cooling (see spectrum after 12 h). Hence, the NTD is able to actively manipulate the equilibrium of RNA foldamers. To overcome the indirect readout of RNA-observed experiments, we recorded protein HSQCs of NTD in complex with Ext along the same temperature steps (Supplementary Fig. 15c). We found a significantly more pronounced complex fraction of NTD-Ext after incubating the sample at 310 K. This effect is particularly well expressed at the sites of the reporter amides used above. In sum, our data support the hypothesis that complexes of

NTD with preferred 5'ge RNAs of SCoV-2 are subject to remodeling at neuralgic sites of the genome.

To confirm, we quantified a temperature-induced increase in complex formation between NTD and RNAs. aSEC at 277 K and room temperature (RT) revealed significantly increased RNP amounts for SL2 + 3 and Ext, and to a lower extent also SL4ext (Supplementary Fig. 15d, e) with an increase in temperature. We concluded that the fraction of NTD-SL4ext is already high at 277 K indicating the genomic context is a NTD-preferred target over Ext alone. At neither temperature we found RNPs of NTD with SL4 or the non-viral ssRNA ss19T, confirming the NTD target preference as independent of temperature.

To verify specific Ext-binding by the NTD also in the SL4ext context we used [15]N-labeled RNA and recorded 2D HN-correlation spectra at 298 K and 278 K (Supplementary Fig. 16a). In line with 1D spectra, we found no signals of Ext at RT, while effects of NTD-binding primarily mapped to the basal stem region of SL4. We conclude this region specifically senses the binding of NTD to the 3'-protruding Ext part. At 278 K, we found exclusively Ext resonances line-broadened with NTD. This supports observations of isolated Ext in 1Ds and the selective interaction of NTD with Ext in SL4ext. In both titrations, we found few residues of SL4 affected. While weak binding fully supports EMSA- and ITC-derived affinities (Fig. 1, Supplementary Fig. 4), we interrogated possible binding sites and titrated NTD to SL4 in RNA 1D spectra (Supplementary Fig. 16b). In accordance with our hypotheses, interactions mapped to SL4 bulges or loops. We thus suggest that NTD exhibits a weak potential to target non-base pairing sections within otherwise structured RNAs, which is supported by previous work[8].

Similarly, NTD-SL3 complexes were indirectly observed by imino peaks in the SL2 + 3 context. Binding to the unfolded SL3 part leads to a stable complex and a reasonable increase in molecular weight of the RNP, which is reflected in significant line-broadening of the SL2 imino peaks (Fig. 7c). In addition, we suggest that analogously to our data on SL4ext (Supplementary Fig. 16), charge-driven, (in)direct interactions with NTD lead to shifting and new SL2 peaks. Here, the effect clearly underlines the particular interdependence of SL2 and 3, also expressed by their seamless transition from one element into the other (Fig. 6c) and supporting the existence of SL2 + 3 as a regulatory hub. Altogether, our data resolve in-detail the preference of the NTD for ssRNA, albeit with clear and fine-tuned discrimination of target RNAs in a particular context.

## Discussion

The multi-modular N protein is highly conserved among coronaviruses indicating its crucial roles related to RNA-binding. How N is capable of combining selective binding, e.g. during transcription, and broad coating of the gRNA[17] during packaging has remained enigmatic. Literature suggests the N IDRs to drive affinity and modulate N's ability to form condensates with RNA[18,22,28,55]. While valuable for a holistic understanding, we lack insight into what exactly triggers N's engagement with gRNA through its designated RBD – the NTD – coordinating time, space, and stoichiometry.

We here used EMSA screening, NMR spectroscopy and biophysical analyses to unravel hallmarks of preferred interactions of N's NTD with SCoV-2 5'ge cis-regulatory SLs 1 to 6. The RNA elements vary strongly in size, sequence and fold and cover stable SLs, AU-rich labile elements, bulges and large, branched elements. As such, they provide a valuable collection of representative viral RNA elements to allow detailed insight into the differential binding behavior of NTD.

Within the 5'-UTR, we identified two elements that correlate with specific signatures of NTD in NMR-observed binding and elevated complex stability, SL2 + 3 and Ext (Fig. 8). NTD's affinity for the elements is in a typical range for specific RBD-RNA complexes[56]. Within the SL2 + 3 region, specific TRS-binding by NTD is established for coronaviruses[37].

Our data unambiguously support the TRS as a preferred NTD target in line with previous studies[8,43]. In addition, we show the Ext RNA is bound by NTD, supporting earlier work[17]. Ext bridges SL4 and SL5, two central SCoV(−2) cis elements. We hypothesize the superior recognition of TRS and Ext is favorably influenced by stable proximal structures like SL2 and SL4. This underlines the important role of genomic context in the recognition by N and may involve other types of RNA regions flanking the actual NTD target site.

Our SAXS and CD data show that both SL3 and Ext exist as transiently folded elements. We suggest their characteristic dynamic behavior adds to preferred targetability. The unique combination of RNA sequence and labile fold could be a particular feature for the virus to recognize its own RNA in the large pool of available motifs (Fig. 8). Likewise, the transient masking of ssRNA might prohibit the binding of those elements by canonical host RBDs, the majority of which also favors ssRNA motifs[56,57]. Of note, Ext not only shows characteristics of a dynamic element in this work, but was previously also found as an integral part of a potential upstream ORF together with SL4[58]. This sheds a new light on the regulatory hub SL4ext and its preferred recognition by NTD.

N's IDRs are capable of interfering with structured RNAs to a much higher, but less specific degree[59]. This allows speculating whether IDRs and NTD act in concert to engage with RNAs. Although discussed earlier[25,43] our data do not support an active role of NTD in RNA-unfolding. Instead, we find NTD to shift ratios of Ext conformers towards ssRNA. N might mask regulatory RNA elements from accessibility to other RBPs and thus e.g. prepare the RNA for packaging. The dynamic interaction of identical cis RNAs in different conformers with multiple RBPs was shown for eukaryotic 3'-UTRs recently[60,61]. For NTD - Ext, our data also stress the crucial role of temperature for viral fitness, which will shift equilibria of neuralgic RNAs to favored engagement with N.

We here established the NTD as to express RNA preferences, which is supported by current literature: i) N was found to appear in multiple proteoforms[62]. In line with our findings, the study suggests the specificity of N is mediated via its N-terminus. Via truncations, N may disconnect specific from non-specific RNA-binding during the viral life cycle. ii) The NTD was shown to be modulated in RNA-binding through arginine methylation at positions 95 and 177[39], both strongly affected in our NMR-based analysis. Selective methylation of arginines will steer RNA-binding at relevant infected-cell stages. iii) The modification of viral RNA will strongly impact specific recognition by NTD as it leads to RNA restructuring[63]. Burgess et al. found excessive m[6]A methylation of the SCoV-2 genome relevant for viral replication[64]. Interestingly, the 5'-UTR is majorly excluded indicating a role of this region for specific RNPs with unmodified sequences. This assumption is corroborated by our data on SL2 + 3 and Ext, exhibiting sequence-encoded intrinsic dynamics, which are less compatible with regulation via methylation. iv) Although the NTD does not invoke LLPS on its own[18,19,65], it was shown to have a measurable influence on condensation[16,17,20]. The different facets of condensates in vitro are suggested to represent the discrimination of capsid assembly and genome processing via phosphorylation of the NTD-flanking SR-region[16]. Potentially, phosphorylation is used to regulate the degree of RNA specificity of NTD during the infectious cycle.

Finally, recent work shows a pivotal role for CTD-mediated dimers of N[18]. N forms structurally dynamic dimers[66] that get compacted upon RNA-interaction and the NTD gets packed onto the CTD[67]. The protein will exhibit higher specificity by facilitating simultaneous interaction with e.g. SL3 and Ext. In that regard, the role of RNA structure for compactness in capsids was shown in a Model Icosahedral Virus[68], possibly in order to position target motifs for N domains in an efficient spatial arrangement, and also suggested as similar requirement in SCoV-2[69].

Given its high abundance and immunogenicity N is a prime target for vaccination and inhibitor search[9,70]. The possible repression of viral replication on the level of LLPS interference[71] has already been

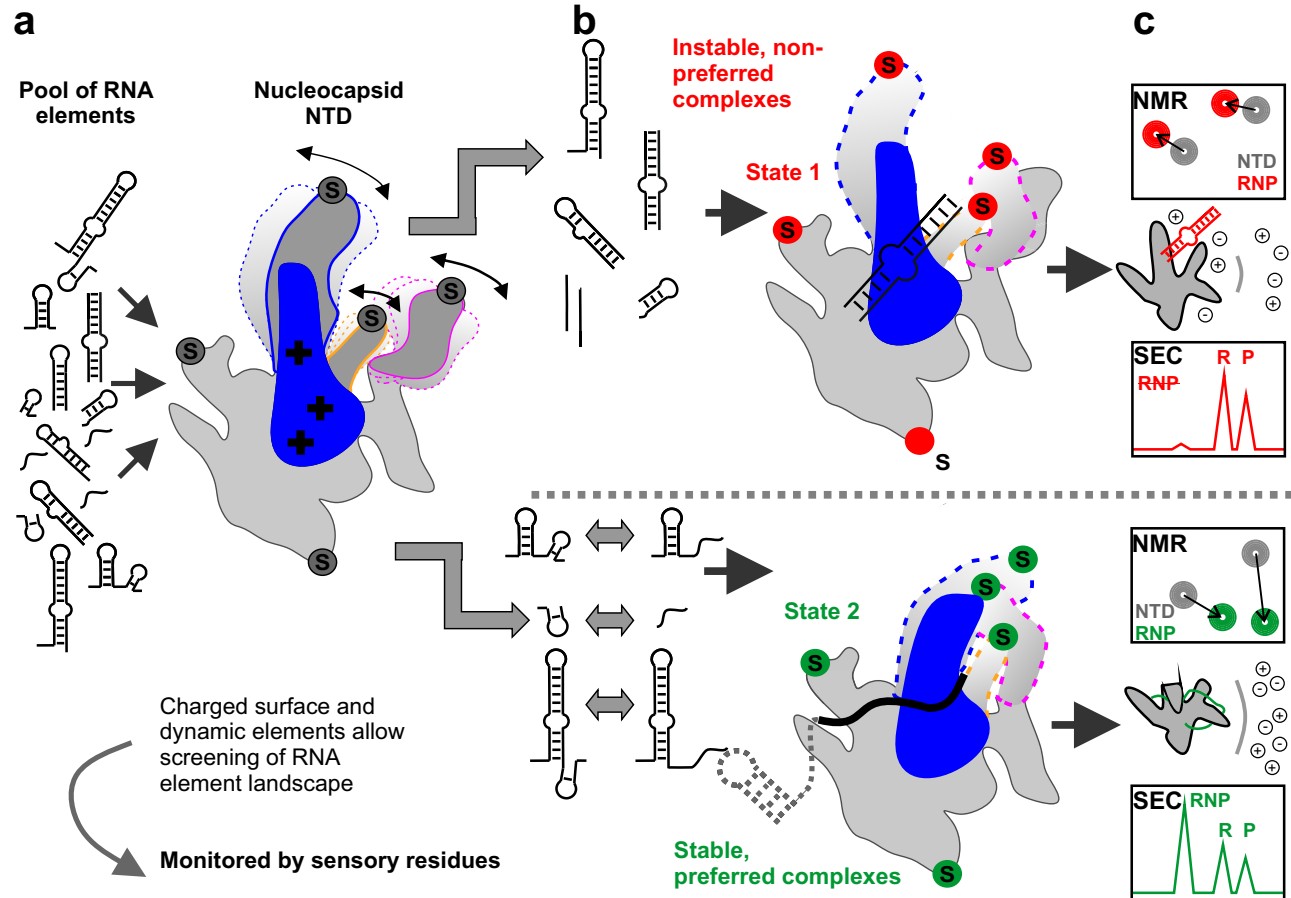

**Fig. 8 | Summarizing model of the NTD's preferential target RNA selection mode. a** The N NTD shows promiscuous binding to a large variety of candidate RNA elements, reasoned by its large positively charged surface that functions like the palm of a hand, together with the central β-hairpin, the basic finger. **b** Dynamic fingers are used to sense (tips labeled with 'S') preferred RNA elements that will lead to stable complexes with the NTD. The NTD not only disfavors pure dsRNA and stable SL elements, but also irrelevant ssRNAs (red). In contrast, preferred RNAs comprise AU-enriched sequences that are transiently masked within labile SLs with melting temperatures around the SARS-CoV-2 optimum (green). The entropic and enthalpic effort for selected RNAs to be present as targetable for the NTD may serve as a proof-reading mechanism for the virus in opposite to canonical ssRNA-binding domains of the host. Possibly, instable NTD-RNA complexes are merely charge-driven, while preferred RNPs might involve the fingers to rearrange into an optimum NTD-RNA conformation. **c** The fingers contain sensory residues, which can report about specific interactions, e.g. via characteristic NMR CSP signatures. Preferred, physiological complexes (RNP) are of low-μM affinities, but reveal physical stability, as e.g. seen in co-elution during aSEC runs and higher salt resistance, as compared to non-preferably bound RNAs. The herein proposed reporter residues can be exploited as a tool to predict the formation of preferred complexes (while the final mechanism remains to be unraveled).

addressed, including the identification of inhibitors, likely against the NTD. Similarly, the herein investigated 5'ge RNA elements were shown to have the principal potency as SCoV-2 RNA drug target sites[42,72]. Our HSQC-based analysis of the NTD binding to 5'-UTR elements now allows for a site-specific interrogation of RNP-druggability using the preferred RNAs.

Finally, different from transcriptional regulation it appears unrealistic for N to use only a few specific sites for genome packaging, which is supported by multiple studies, albeit in parts controversially[4,17,54]. Our data provide insights into the subtle differences in RNA motifs that likely account for specific recognition by the NTD. A recent study has presented SCoV-2 virus-like particles that allow examining the precise role of N sequences and RNA motifs for quantifiable packaging[54]. The detailed meaning of particular NTD complexes with RNA motifs in a natural context, e.g., of such studies, can now directly be correlated with atomic signatures.

## Methods
### Experimental temperatures
Exact temperatures are given in Kelvin for each experiment, either in the Methods Section or – if required – in the corresponding Fig.

legend. Whenever experiments were carried out without distinct temperature control, we refer to it as room temperature (RT). By our definition – and to our best knowledge – RT spans a range between 22 and 25 °C.

### Construct design
The SCoV-2 N protein NTD coding sequence was based on NCBI reference genome entry NC_045512.2, identical to GenBank entry MN90894[1]. Domain boundaries were defined in analogy to the available NMR structure (Protein databank, PDB: 6YI3)[8], spanning amino acids 44 to 180. The NTD wildtype (WT) and point mutants (G60I, G99I, S105I and R107A) were cloned as described in the Supplementary Methods.

### Protein production
Production and purification of the SCoV-2 N NTD and mutants thereof were guided by[8] and recently described in detail by us[73]. Briefly, uniformly ($^{13}$C and) $^{15}$N-labeled (or unlabeled) NTD protein was purified via size exclusion chromatography on a Superdex™ 75 HiLoad 16/600 column (Cytiva) in NTD standard buffer (25 mM KPi pH 6.5, 150 mM KCl). The concentrated NTD protein was further subjected to ion

exchange chromatography to remove possible traces of co-purified RNases in the sample. To this end, the salt concentration of the NTD protein sample was adjusted to 50 mM KCl before applying it to a 1 mL ENrich™ S (Bio-Rad) cation exchange column equilibrated with 25 mM KPi pH 6.5, and 50 mM KCl. A continuous KCl gradient from 50 mM to 500 mM was used to elute the NTD at an approximate KCl concentration of 100 mM. The salt concentration of the final sample was adjusted to 150 mM KCl prior to further use. 2 mM of TCEP or DTT was added for samples to be used in SAXS and MALS.

## NMR

All software applications related to structural biology, including structure visualization and image creation by PyMol (Schrödinger) have been run via the SBGrid platform[74]. NMR measurements were carried out at the Frankfurt BMRZ using Bruker spectrometers of 600–950 MHz proton Larmor frequency, equipped with cryogenic probes and using Z-axis pulsed field gradients. All RNAs, NTD, and complexes were consistently measured in NTD buffer including 5% v/v $D_2O$ at 298 K or as otherwise indicated in figures or at relevant text sites. Data acquisition and processing were undertaken using Topspin versions 3 and 4. Cosine-squared window functions were applied for apodization in all dimensions. Spectra were referenced with respect to added DSS and for $^{13}C/^{15}N$ as suggested in[75]. Analysis of CSPs and het-NOE ratios was performed using the CCPNMR analysis 2.4 software suite[76] and the program Sparky[77]. Details are given in the Supplementary information.

The {$^1H$}$^{15}N$ heteronuclear NOE experiments were performed as interleaved HSQC-based pseudo-3D versions including solvent suppression by WATERGATE sequence[78] and a saturation delay of 6 s from samples of 160–340 μM (apo) and 220 μM in complex with Ext. Guided by a previous study on SARS-CoV NTD[44] we recorded hetNOE experiments of the protein at various field strengths (298 K each, proton Larmor frequencies of 600 MHz, 800 MHz, and 900 MHz).

$^1H$-$^{15}N$-HSQC experiments were acquired with a nitrogen offset at 117 ppm and a constant spectral width of 34 ppm using 96–256 indirect complex points. For titrations, we used 128 indirect points and complex forward linear prediction until 192 points. For RNA titrations, we added RNA to the apo NTD sample after running the reference HSQC to the final titration point and subsequently re-mixed sub-stoichiometric ratios with fresh apoprotein from the identical batch. For RNA-observed titrations, an increased number of scans (see Source Data) compensated marginal dilution. For salt titrations of NTD in complex with 1.2x molar excess of 5'ge RNA elements $^1H$-$^{15}N$-HSQC were recorded after adding respective volumes of a 3.3 M KCl stock solution, with subsequent incubation for four hours to allow samples to reach equilibrium at 50, 140, 240, and 405 mM, respectively. Spectra of complexes and the NTD alone were obtained from measurements at 800 MHz, 298 K, using 32 scans per FID. Combined $^1H$-$^{15}N$-chemical shift perturbations (CSP) were calculated in ppm according to eq 1

$$CSP = \sqrt{\left(\frac{\delta N}{5}\right)^2 + (\delta H)^2} \qquad (1)$$

Significance was defined as the mean plus standard deviation.

Imino proton measurement of RNAs was carried out via jump-return 1D spectra[79] applying a binomial water suppression delay optimized for the center of respective imino proton resonances.

Control spectra of RNA alone for the salt titrations were acquired at 800 MHz, 298 K, 3072 direct FID points, using 256 scans, and a water recycle delay of 39.06 μs.

For the BEST-TROSY-based titration of $^{15}N$ SL4ext with NTD we used a sample of 120 μM in a 5-mm tube and acquired 64 scans with a recycle delay of 0.3 s. Unlabeled NTD was added at 298 K stepwise from a stock of 532 μM to a maximum of twofold excess, and the

number of scans was adjusted to compensate for dilution. Complex spectra were sequentially measured at 298 K and 278 K.

$^{15}N$ relaxation data of NTD alone and in complex with Ext RNA were acquired as pseudo 3D-experiments at 298 K and a field strength of 600 MHz including temperature compensation. We used the following T1-delays: 10 ms, 30 ms, 50 ms, 90 ms, 150 ms, 250 ms, 500 ms, 1000 ms, and 1500 ms (NTD-Ext) or 2000 ms (apoNTD). For T2, we used delays of 16.96 ms, 33.92 ms, 50.88 ms, 67.84 ms, 101.76 ms, 135.68 ms, 169.6 ms, 203.52 ms, and 271.36 ms for the apo NTD and 16.96 ms, 33.92 ms, 50.88 ms, 67.84 ms, 84.8 ms, 101.76 ms, 118.72 ms, 135.68 ms, and 169.6 ms for the NTD-Ext complex. All recycling/pre-scan delays were set to 2 s. Data were analyzed and fitted with the CCPNMR analysis 2.4 software suite[76], with the given errors being a measure of the fit quality.

## SAXS

In-detail procedures on SAXS measurements and relevant information on the utilization of derived values are given in the Supplementary information. Briefly, SEC-SAXS measurements were performed at 293 K at the EMBL P12 BioSAXS beamline at PETRAIII (DESY synchrotron Hamburg)[80] equipped with a Pilatus 6 M 2D photon counting area detector. All samples were mixed from purified components, shock-frozen in liquid nitrogen, delivered to the beam line, defrozen and subjected to SEC-SAXS after extensive centrifugation. Prior testing revealed that shock-freezing and thawing does not negatively influence sample states. Samples were normally delivered 1-2 days before beamtime and locally stored at −80 °C. For SEC-SAXS runs, 90 μL of protein, RNA, and protein-RNA complex samples ( ~ 300 μM each) were loaded onto a Superdex75 Increase 10/300GL (GE Healthcare) in standard phosphate buffer using an Agilent 1260 Infinity Bio-Inert HPLC system at a flow rate of 0.6-0.7 mL/min. Between 2000–2520 successive 2D SAXS data frames of 0.995 s each were collected from the continuously flowing eluate spanning one column volume (24 mL) and delivered directly to the beam line.

The following data averaging and reduction was undertaken using the ATSAS 3.0 software suite[81] (detailed in the Supplementary Methods): After the subtraction of appropriate background/buffer scattering contributions from the SEC-peak sample frames, we used the CHROMIXS[82] -derived $R_g$ values calculated through the SEC-elution peaks, to guide the scaling and averaging of the SAXS data and produce the final 1D profiles as reported in the text (Fig. 4, Supplementary Fig. 8 and also see Supplementary Table 2). The Supplementary Methods precisely describe the selection of frames for samples and buffers. The final $P(r)$ curves in Fig. 4 are normalized to the forward scattering intensity, $I(0)$, using the BIOXTAS RAW suite[83]. All final curves were processed to derive $R_g$, Porod volumes, $P(r)$ and molecular weight estimates using Bayesian inference[84]. Structural parameters and relevant quality of fit assessments are reported in Supplementary Table 2 as guided by the recommendations of Trewhella et al.[85].

We used the finally processed SEC-SAXS data to create structural models for SL2 + 3, SL4, Ext, and SL4ext with RNAmasonry[51] via 50 iterative steps using CRYSOL[86] as a model fit procedure. Secondary structure was manually given for the initial step, based on the models as described in Supplementary Fig. 1 and kept unrestrained for folding and 3D-model building. Final models underwent manual protonation using PyMol (Delano Scientific, Schrödinger). The NTD X-ray crystal structure fit to the SAXS data (PBD: 6M3M) was evaluated using CRYSOL. All SAXS data and relevant models are available in the Small Angle Scattering Biological Databank[87].

## RNAs

We used RNA secondary structure models of 5'ge RNAs SL1-SL6 taken from recently published data[30,31]. For SL2 + 3 (5' and 3') and SL4ext (5'), we used slightly extended constructs to allow for native boundaries of stem-loops (Supplementary Fig. 1). Unlabeled RNA constructs used in

this study were produced by in-house optimized in vitro transcription and purified as follows: Plasmid-DNA[31] was linearized with *Hind*III prior to in vitro transcription by in house expressed T7 RNA polymerase. Preparative-scale (10 to 20 mL) transcription reactions (4 h at 37 °C) were terminated by the addition of ethylenediaminetetraacetic acid (EDTA) and the RNAs were precipitated with 2-propanol overnight at −20 °C. RNAs were separated on 12–16 % denaturing polyacrylamide gels and visualized by UV shadowing. Excised RNA-fragments were eluted into 0.3 M NaOAc overnight and subsequently washed, concentrated, and buffer-exchanged to NTD buffer. $^{15}$N-labeled RNA (SL4ext) was produced accordingly, with $^{15}$N-labeled uridine (rUTP) and guanosine (rGTP) (Cambridge Isotope Laboratories), and unlabeled adenosine (rATP) and cytidine (rCTP) (Sigma-Aldrich). Sequences of SCoV-2 5′ge RNA elements and of CoV-non-related RNAs are listed in Table 1 and detailed in Supplementary Table 1. HDV ribozyme coupled DNA templates for the 5′ge elements SL1, SL2 + 3, SL4, SL5, and SL6 were kindly provided by the *Covid19-nmr* consortium. SL2 + 3, SL4ext and Ext DNA templates were kind gifts of Julia Weigand's lab (Marburg University), cloned by Stephen Peter. For P2, ss19T, ss19B, and SL_AUA, complementary oligonucleotides (Sigma Aldrich) were annealed and cloned into an HDV-containing vector using Gibson assembly (Gibson et al., 2009). For P3, P3_A, P3_B, SL1_ext$^{SL1}$, SL1_ext$^{SL4}$, SL4_ext$^{SL1}$, complementary oligonucleotides (Sigma Aldrich) were annealed and used as templates for in vitro transcription. Final RNA samples were buffer-exchanged to NTD buffer and sample quality, homogeneity, and long-term stability were verified by native and denaturing PAGE as well as 1D NMR experiments by means of the characteristic imino proton pattern. Details are given in the Supplementary information.

### Analytical size-exclusion chromatography (aSEC)
Analytical SEC runs at RT were performed by loading 100 μg of protein, protein-RNA complex (100 μg protein with a 1.2-fold molar excess of RNA), or RNA (varying RNA amounts corresponding to 1.2-fold molar excess as in the complex) samples, respectively, onto a Superdex75 Increase 10/300GL column (GE Healthcare) equilibrated with NTD buffer using an ÄKTA-explorer FPLC. Flow rates were set to 0.75 mL/min. The run was monitored with two UV absorbance (260 and 280 nm) traces. All SEC runs were analyzed using ChromLab_v6 (Bio-Rad) and the traces were plotted in OriginPro. Additional details are given in the Supplementary information.

### Melting temperature ($T_M$) analysis
The UV-observed melting of various SCoV-2 5′ge RNAs used in this study was carried out on a JASCO J-810 spectropolarimeter equipped with a Peltier temperature control module (JASCO). 20 μM RNA sample in NTD standard buffer was used for measurements in a cuvette (Helma QS) with a sample length of 1 mm. Both CD and UV absorbance were monitored. RNA melting experiments were recorded at 260 nm from 5 to 95 °C with a heating rate of 1 °C/min. The bandwidth was set to 1 nm with a digital integration time of 1 s. Melting temperatures were obtained by fitting normalized raw UV absorbance data with a one- or two-transition model in OriginPro. The first derivative of the normalized UV absorbance at 260 nm over temperature was plotted to derive transition points.

### Electromobility shift assays (EMSAs)
For the initial analysis of NTD RNA-binding preferences, radioactive EMSAs were performed according to reference[88] with the following modifications: 30 pmol RNA transcripts were dephosphorylated using Quick CIP (NEB) following the manufacturer's protocol and resuspended in $H_2O$. Subsequently, 5′ end-labeling of 15 pmol RNA transcripts with [γ-$^{32}$P]-ATP was accomplished with T4 polynucleotide kinase (NEB). Labeled RNA was separated from unincorporated [γ-$^{32}$P]-ATP by column purification (NucAway) and adjusted with NTD buffer

(25 mM potassium phosphate, 150 mM potassium chloride, pH 6.5) to 0.03 pmol/μl. Binding was performed for 10 min at RT in 20 μl reaction volume in the presence of 0.6 μg tRNA from baker's yeast (Sigma), 3 nM $^{32}$P-labeled RNA, 1 mM MgCl$_2$ and various dilutions of NTD in NTD buffer. After the addition of 3 μl loading buffer (30% glycerol, bromphenol blue, xylene cyanol) the RNP complexes were resolved by PAGE (6% polyacrylamide, 5% glycerol, and 1×TBE) at 80 V for 75 min at 23 °C, with pre-cooled (4 °C) TB running buffer (0.13 M Tris, 45 mM boric acid) for improved resolution. Gels were dried and subsequently exposed to a phosphor imager screen and visualized using a Typhoon laser scanner (GE). Images were exported using ImageQuant TL (v8.1).

### Reporting summary
Further information on research design is available in the Nature Portfolio Reporting Summary linked to this article.

## Data availability
All SAXS data in phosphate buffer generated in this study have been deposited in the SASBDB under the following accession codes: SASDPK6 (NTD); SASDPL6 (SL2 + 3); SASDPM6 (SL4); SASDPN6 (SL4ext); SASDPP6 (Ext); SASDPQ6 (NTD: SL2 + 3); SASDPR6 (NTD: SL4ext); SASDPS6 (NTD: Ext); SASDPT6 (NTD: SL4). All SEC-SAXS-MALS data in HEPES have been deposited in the SASBDB under the following accession codes: SASDR33 (NTD); SASDR43 (SL2 + 3); SASDR53 (SL4); SASDR63 (SL4ext); SASDR73 (Ext). NMR spectral resonance assignments of this study use the following previously published entries for NTD and RNAs in the BMRB: BMRB 34511 https://doi.org/10.13018/BMR34511 (NTD); BMRB 50654 https://doi.org/10.13018/BMR50654 (5_SL2 + 3); BMRB 50657 https://doi.org/10.13018/BMR50657 (5_SL4). Buffer-related minor changes in NMR chemical shifts or in RNA-derivatives are shown in the manuscript figures and the underlying processed spectra generated in this study are provided in the Supplementary Information together with assignments. Additionally, the new imino proton assignments for Ext alone and the full imino group assignments of SL4ext have been deposited in the BMRB under the following accession numbers: BMRB 51955 (Ext imino protons); BMRB 51956 (SL4ext imino groups). All NMR spectra underlying the herein-presented data will be provided upon request. NTD structures used within this study are available PDB entries under the following accession codes: 6YI3 (NMR structure); 6M3M (crystal structure). Material requests shall be made to the corresponding author. According to the open-source policies of the *Covid19-nmr* consortium, all RNA- and protein-production constructs are available upon request. All other data are available from the corresponding author on request. Source data are provided with this paper.

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

## Acknowledgements

The Frankfurt BMRZ acknowledges support from the state of Hesse. This work was supported by the Goethe Corona Funds, by the German Research Council (DFG) within CRC902 ("Molecular Principles of RNA-based regulation") project part B18, through DFG grant number SCHL2062/2-1 to A.S., and by the Johanna Quandt Young Academy at Goethe through the financial support of A.S. (stipend number 2019/AS01). The authors acknowledge the in-kind support of the DESY Strategy Fund Corona-related research project scheme (200702, to CMJ). We acknowledge the *Covid19-nmr* consortium for providing plasmids for previously published RNAs, and Anna Wacker for the excellent coordination of RNA-related data exchange, lab standards, and comprehensive input. We thank the Weigand and Schwalbe labs (Frankfurt and Marburg Universities) for kindly providing plasmids for SL2 + 3, Ext, and SL4ext; and Anna Wacker, Christian Richter, Melina Müller, Matthias Becker, and

Harald Schwalbe for providing RNA spectra and fruitful discussions on NMR analyzes of RNAs. We thank the BioSAXS team at EMBL Hamburg P12 beamline for the valuable discussion. Katharina Targaczewski is acknowledged for excellent technical wet lab support.

## Author contributions

S.M.K. and A.S. initiated the project. S.M.K. and K.D. carried out protein production and sample preparation for SAXS, NMR, ITC, and CD measurements. S.M.K. and A.S. performed NMR experiments. K.D. performed SAXS and CD measurements. C.M.J. and K.D. carried out data processing of SAXS and MALS and carried out data deposition at SASBDB. S.M.K. and A.S. performed the NMR assignment, chemical shift perturbation analysis and IVT of RNA constructs. All four authors analyzed and discussed data. S.M.K. and A.S. wrote the manuscript.

## Funding

## Competing interests

The authors declare no competing interests.
