## [Peer Review File · Nature Communications]

REVIEWER COMMENTS

Reviewer #1 (Remarks to the Author):

In this manuscript, authors are investigating the binding of the N-terminal RNA binding domain (NTD) of the SARS-CoV-2 nucleocapsid protein to conserved and structured RNA elements located in the 5' untranslated region of the viral genome. A large set of biophysical methods (mainly NMR, but also SAXS, ITC, CD and UV spectroscopies...) are used to understand the specificity of binding such as identification of protein residues involved into the interaction or RNA elements that are preferentially recognized. The manuscript is in general well written and illustrated but could be significantly shortened.

Below are a several comments on some of the issues that need to be clarified:

- Line 119 "We found micromolar affinities"; data derived from EMSA are only apparent affinities.
- In the Supplementary Fig. 5 showing SEC data, two peaks are visible upon NTD addition to the SL4ext RNA construct at 4°C (instead of one peak for all other RNA-protein complex). This should be commented.
- Assignments of imino protons are missing in Fig. 6b, 6c and Supplementary Fig. 8b, 9c, 10a. This should be added. It would be useful to have the assignment indicated in the figures, as well as one supplementary figure showing detailed secondary structures of all RNA studied in the article.
- Page 12, lines 375-378 and Supplementary Fig. 9e about ITC data: the fit presented is by far not of an acceptable quality. As a result, the affinities presented are not significant (and in any case cannot be compared with apparent affinities obtained by EMSA at an unknown temperature). A table showing thermodynamic parameters derived from ITC data is missing. ITC data were unfortunately collected only for the SL4ext-NTD interaction (they should be repeated at least for the SL4 RNA construct) and fitted using a two sites binding model which I doubt is suitable for the proposed binding mechanism: this model would implies that about 50% of the NTD binds the RNA in some way (Kd1, DH1, DS1) and about 50% binds in a different way and/or on a different binding site (Kd1, DH1, DS1). ITC data could be highly valuable for this study and would deserve more attention.

- The authors did not give any details regarding titrations of RNA by addition of increasing amount of NTD. This information is missing in the Materials and Methods section. The dilution of the sample due to the addition of the protein is not indicated. This is especially important for experiments depicted in Fig. 7b and 7c. Page 13, lines 412-413, the authors discussed a “stepwise decrease of imino signals” and concluded that this suggests that the NTD interferes with the equilibrium of the Ext conformation. The decrease of imino proton signal is very weak (Fig.7b) and without other elements and details about the way the titration was performed, this conclusion cannot be stated. In Fig. 7c, and page 13, the authors described “line-broadening of the SL2 imino peaks”. However, analysis of the titration clearly shows appearance of new peaks, disappearances, shifts and broadening of resonances. This has to be described. It is not only a line-broadening phenomenon.

- In the Supplementary Fig. 9c, spectra of SL4 and SL4ext are shown at 278K, 298K and 310K. In the Supplementary Fig. 9d, a minor CS differences is shown but the temperature is not indicated in the legend (should be 298K I guess ?). Below 298K, spectra exhibit some significant differences and this should be discussed. At 298K, the authors mentioned that the Ext region does not interfere with the folding the SL4 domain but, below this temperature, this should be verified, especially since the titration for Ext was performed at 15°C. The imino protons of SL4ext should be assigned at low temperature by analysis of NOESY and 1H-15N HSQC.

- In the Supplementary Fig.12, assignments are missing: G85, U89, U120, U115 and G102 should be added. The temperature should be added in the legend. It is surprising that the resonance of g-1 appears with a high intensity. The imino proton of the first base pair in RNA is often broaden due to fast exchange with the solvent. Could authors comment on this? It should also be explained why only imino proton adjacent to bulge or loop undergo line-broadening without altering the rest of the structure, in particular the adjacent base-pairs? Finally, new peaks are showing up at 10.7 ppm: do they arise from the protein or the RNA?

- The conclusion and discussion describing the interaction between RNA and the NTD protein suffer from an important missing experiment: the titration of SL4ext with increasing concentration of NTD, as shown in the case of SL4 in Fig. S12. This titration should be performed at 15°C by monitoring imino proton region using one-dimensional and 1H-15N HSQC experiments. The SL4ext should be 15N labeled and assigned as described in the Supplementary Fig. 12 in the case of SL4. In the Fig. 1, EMSA experiments are depicted and the NTD shows a stronger affinity for SL4ext in comparison to Ext and SL4. In the Fig.2, the CSP analysis of NTD binding to RNA shows that the SL4ext exhibits the highest values. All these experiments suggest a cooperativity between Ext and SL4 domains for binding to the NTD. As imino protons are visible at low temperature, the titration of SL4ext should bring important details about the interaction and the respective roles of SL4 and Ext domains.

- Reference #39 should be updated (now published in a journal).

In conclusion, although the work is original in the field, several aspects should be seriously improved. Its significance could be strongly improved by obtaining a structure for the stable SL4ext-NTD complex.

Reviewer #2 (Remarks to the Author):

The article by Korn and colleagues is a detailed investigation of the RNA-binding properties of the N-terminal region of the N protein from SARS-CoV-2, using ligands derived from its genomic RNA. The N protein contains both specific and non-specific RNA-binding properties, and yet still functions in a precise manner during the viral cycle. As such, its RNA recognition properties are complex. The authors have done a good job to provide insight into what may define the specific elements of the interactions despite an apparent high degree of variation in the known N protein binding hot spots in the genomic RNA. I find that the series of experiments are performed well, and encompass a wide range of biophysical approaches to study the topic. This study is an important contribution to the understanding of the N protein function. I have no major concerns regarding the work, but nonetheless I have some questions to be addressed:

- it would be great to include a supplementary graphical figure to go with the ligands in Table 1. It takes a bit of effort to determine exactly where the constructs start and stop within the genomic RNA structural elements. This also relates to the next point.

- given that N protein has been linked to binding ssRNA in the past (such as on line 127) would it not have made sense at some point to extend one or more ligands to include the ssRNA parts before or after the stemloop(s)? This is partly addressed in the final experiments (such as Suppl. Fig 10b,c) but could be more explicitly referenced in the manuscript. I am not suggesting a requirement for new experiments (although perhaps some additional results have already been obtained that addresses this question?). However, I think it should be more clearly stated that the N protein RNA ligands did not include the adjacent ssRNA (even though they may actually be recognized by protein N..)

- I found the careful study of generic ssRNA binding vs specific ssRNA to be particularly useful to understanding the N protein RNA preference. Just out of interest for potential discussion, do you think that in the cell the transient dsRNA-ssRNA segments might also be important hot spots, in that since they are not completely ssRNA, they might be 'protected' from interaction with the plethora of

ssRNA binding proteins in the cell? That is, the partial dsRNA nature would prevent or lessen recognition by many of the host RNA-binding proteins? Therefore N protein could still be specific for ssRNA elements, which would be clear in vitro, but the weak dsRNA/ssRNA character could be critical within the cell.

- a final model would help gather the main findings together. I find that the current final model (Suppl. Figure 13) does not provide much help in summarizing the main results and is perhaps mostly speculative. I would instead suggest to provide a final summary that focuses on the favoured targets of protein N (weak dsRNA that have a high ssRNA population) and any other main aspects of the interaction and mechanism that are derived from the results.

- in the first two paragraphs of the results, there is some discrepancy between the statements that the NTD differentiates by size (line 111) and that the Ext and SL2+3 have similar affinities (line 124). This would also apply to the large, but poor binding SL4. Maybe the size dependent statement can be removed?

- Figure 2 would benefit from a secondary structure diagram above the graphs, just as in Figure 3, to help the reader identify where the CSP regions are in relation to the protein

- in the paragraph from lines 182-187 the importance of the multiple field strength is not clear, nor is the conclusion that the b-hairpin displays a rigid-body dynamic. Perhaps this paragraph can be removed since the key finding is already mostly part of the last sentence in the previous paragraph (lines 179-181).

- interpretation of the dynamics results in the section from 296-328 requires additional clarification or instead a clearer explanation of potential relevance of the observed dynamics and how it relates to RNA binding. For example, what would be an advantage to coupling the observed dynamics to RNA preference.

- although the experiments are well described for the last two sections of the results, I found several places where the text was unclear. For example, lines 401-409 are not clear to mean regarding the main point that is being made - perhaps some words were accidentally removed? Or as another example on line 436, maybe the end of the sentence should be 'with an increase in temperature' or something similar.

- the figure icons for Fig. 8c,d,e are not the easiest to follow. I assume that the number of RNA that are stemloops or curved lines represents a shifting population? However, at present it looks a bit like the RNA is oligomerizing on the protein or some other more complex behaviour. Unfortunately I do not have a specific recommendation for increasing clarity, except perhaps in figure c and d to indicate that protein N NTD shifts the equilibrium towards the ssRNA, maybe by placing equilibrium arrows between a single unfolded and folded RNA, and have protein N beside the arrow from stemloop to ssRNA. And then increase the arrow width at different temperatures to reflect changes in the population. But this is just one possibility.

- for the Discussion, my other suggestion is to remove the Supplementary Fig. 13 and perhaps even the paragraph from 537-547 since it is not that clear in its current form, and is outside of the main theme of the manuscript.

- tiny point: some figure panel letters are missing in the text (such as line 229 should be Suppl. Fig. S6'A'), and in line 199 it is not clear which figure the 'panel a' is referring to.

Reviewer #3 (Remarks to the Author):

This is a potentially very interesting story describing the biophysical characterization of an important protein-nucleic acid complex and there is a good degree of experimental design (lots of different constructs, lots of tests).

However, much more information needs to be provided about the SAXS data before this could even be considered for publication. The authors do not show raw SAXS data in the text, only the result of processing which can (but does not have to) be impacted by parameters selected during the analysis. The lone exception is found in Supplementary Figure 6. It is stated in the text that “the quality of our SAXS data is well reflected in an excellent fit of an experimental NTD scattering curve compared with back-calculated SAXS scattering from an available crystal structure (Supp Fig 6).” To

my eye, Figure 6 shows a (partially) aggregated sample and a rather poor fit at the most important, low q values, which assess the heterogeneity of the sample. The authors do not appear to recognize that this low q signal is a problem, thus raising concerns about the interpretation of all of the rest of their SAXS data. If the NTD is not monomeric, how can association be properly characterized? The following statement in Methods was also concerning: “initial points [were] removed to enable shape fitting”. Does this imply that aggregation was evident?

Additional concerns arise from the statement that “SEC-SAXS allows analysing individual fractions of NTD, RNA and RNPs”, and other references to the use of SEC-SAXS to measure specific fractions. If Figure 6 is the result of a SEC-SAXS experiment, the columns may not have properly separated the fractions. As a result, it is hard to accept the interpretations of the SAXS data, without further proof that the samples are in the monomeric state.

The authors should read and follow the guidelines put forth in: “2017 publication guidelines for structural modelling of small-angle scattering data from biomolecules in solution: an update”, *Acta Cryst.*, D73, 710-728 (2017). Please use the table in that document to provide information about model independent parameters to allow a fair assessment of the SAXS data. And please show the raw data.

Other comments:

I disagree that the changes in Figure 4a shown for SL2+3 are any more (or less) significant than those shown for SL4. The D_{\max} is a ‘selected’ parameter, and given the above raised concerns about aggregation, it needs to be double checked. Please show the raw profiles, the answers will be obvious from looking at the curves.

The computed MW for Ext is inconsistent with the quoted R_g , and the asterisked comment that might explain the difference does not appear in Table 2. In fact, the R_g suggests that the sample is aggregated, in conflict with the MW. What are the error bars?

The sample concentrations for the SAXS data were never reported. Concentration is critically important for SAXS experiments.

The authors, in several places, comment on the ‘excellent’ work performed by others. This seems unusual.

Reviewer #4 (Remarks to the Author):

Korn et al describe “the preference signature of SARS-Cov2 nucleocapsid NDT for its 5’ genomic elements”. I find the manuscript difficult to read and I don’t see the bigger picture/ general interest for this work. If we want to understand the binding preference of nucleocapsid NDT than learning how it binds a handful of structures is not sufficient. Do the authors gain any predictive power? Can they predict whether (or which) the NDT would bind to 5’UTR structures of other coronaviruses ? What about other families of RNA viruses? How does the NTD domain discriminate between RNA stem-loops from coronavirus verses stem loops within cellular mRNAs? The main claim of the abstract “We show that the domain uses a set of flexible sensory residues to read the intrinsic signature of preferred RNA elements for selective and stable complex formation within the large pool of available motif” is not supported by structure/functional analysis – the authors should mutate some of this “sensory” residues and predict the outcome of these mutations in terms of RNA structure recognition and binding. Overall, the authors perform a lot of detailed biophysical characterization of the NDT but lack testable predictions and are somewhat repetitive such as some of the figures seem better suited for the supplementary. For example, it wasn’t clear to me is Fig 3c entirely consistent with the gel shifts from figure 1D? Each figure should emphasize the critical new information that is being added. In my opinion this is careful and rigorous work but without

mutational characterization or deriving general principles for discrimination of structure motifs, this work is better suited for a more specific journal.

We thank all four reviewers for the constructive suggestions on additional experimental strategies and included additional data to improve its quality. We thus hope to meet all of the raised concerns. In line with the reviewers' suggestions, we significantly shortened the manuscript, especially by reducing redundancies. Facing an overall increase in data we have uploaded a Source Data file for replicates, raw data, spectra etc. and thus created space to rearrange figure panels and include new ones.

Note that all significant text changes as well as new paragraphs in the manuscript are marked in red submitted together with the revised clean version. For a comprehensive overview of text compaction, see attached document "Summary of changes".

Reviewer #1 (Remarks to the Author):

In this manuscript, authors are investigating the binding of the N-terminal RNA binding domain (NTD) of the SARS-CoV-2 nucleocapsid protein to conserved and structured RNA elements located in the 5' untranslated region of the viral genome. A large set of biophysical methods (mainly NMR, but also SAXS, ITC, CD and UV spectroscopies...) are used to understand the specificity of binding such as identification of protein residues involved into the interaction or RNA elements that are preferentially recognized. The manuscript is in general well written and illustrated but could be significantly shortened.

We thank the first reviewer for the overall positive assessment of the manuscript's text and illustrations. We completely agree with the perception that the manuscript has been too long. Despite a large amount of new data, we shortened the manuscript in the individual results sections and the discussion. We summarize this shortening effort in the attached document "Summary of changes" for a comprehensive overview. This file further contains a summary of new and rearranged Figures.

Below are a several comments on some of the issues that need to be clarified:

Line 119 "We found micromolar affinities"; data derived from EMSA are only apparent affinities.

We thank the reviewer for pointing to this inaccuracy. We adjusted it accordingly in the text.

In the Supplementary Fig. 5 showing SEC data, two peaks are visible upon NTD addition to the SL4ext RNA construct at 4°C (instead of one peak for all other RNA-protein complex). This should be commented.

Former Suppl. Fig. 5 → now Suppl. Fig. 6

Indeed, we have ignored this observation previously. We suggest that at non-physiological 4°C the two peaks in complex with the NTD may derive from SL4ext RNA where we find partial opening of the basal SL4 stem, in line with our new findings from ¹⁵N-SL4ext observed titration with NTD (see also response to respective comment below). This possibly leads to a modulated running behavior and two different elution volumes caused by differential hydrodynamic radii for SL4ext conformers. Former Supplementary Fig. 5 is now Suppl. Fig. 6. As we are aware of the lack of a final proof for this we have now included the following statement in the Figure Legend to Suppl. Fig. 6 and Suppl. Fig. 15, to not over-interpret findings:

“Of note, we observed two complex peaks for the SL4ext at low temperature, which likely represents two different conformations of the bound RNA. In accordance with the imino proton assignment spectra at 278K (Supplementary Fig. 11), those conformations might represent 1) the completely base-paired SL4 moiety, and 2) the opened-up lower SL4 stem moiety.”

Assignments of imino protons are missing in Fig. 6b, 6c and Supplementary Fig. 8b, 9c, 10a. This should be added. It would be useful to have the assignment indicated in the figures, as well as one supplementary figure showing detailed secondary structures of all RNA studied in the article.

Former Suppl. Fig. 8 → now Suppl. Fig. 9. Former Suppl. Fig. 9 → now included in main text Fig. 6. Former Suppl. Fig. 10 → now Suppl. Fig. 12.

We have now added all imino proton assignments for the SARS-CoV-2 RNA elements at the requested sites (note the renumbering of figures, listed in attached file “Summary of changes”). We have used this option to more precisely look into iminos of all RNAs in dependence of temperature (while sparing data on 310K to reduce content; see response below) and NTD-binding. In this context we have corrected the underlying SL2+3 construct sequence and expanded the SL2+3 secondary structure scheme of our construct to not drop a second conformation in SL3 at 278K. The interpretation of binding data is not affected, but in fact underlines the dynamic nature of this NTD target region. All assignments of the SL2 - Ext region are summarized and detailed in the new Suppl. Fig. 11.

For the RNA spectra shown in former Suppl. Fig. 10a (now Suppl. Fig. 12a), we would appreciate to spare the process of imino assignments seeing that RNAs are either single-stranded (ss19), double-stranded with the expected number of signals (ds19) or a pure-GC-stemloop (SL_AUA; with 6 out of 7 GCs visible), the latter with a large overlap of chemical shifts. For all RNAs, we provide their 1Ds to confirm their single-stranded or folded nature, which we assume is in line with the spectra shown, based on the number of base pairs. No particular tracking of signals is performed for those RNAs, that would necessitate an assignment.

We have also included an overview figure in the SI (new Suppl. Fig. 1) of all RNAs used, which includes the secondary structure predictions and the genomic numbering according to Table 1 and Suppl. Table 1 (see also comment to reviewer 2).

Page 12, lines 375-378 and Supplementary Fig. 9e about ITC data: the fit presented is by far not of an acceptable quality. As a result, the affinities presented are not significant (and in any case cannot be compared with apparent affinities obtained by EMSA at an unknown temperature). A table showing thermodynamic parameters derived from ITC data is missing. ITC data were unfortunately collected only for the SL4ext-NTD interaction (they should be repeated at least for the SL4 RNA construct) and fitted using a two sites binding model which I doubt is suitable for the proposed binding mechanism: this model would imply that about 50% of the NTD binds the RNA in some way (Kd1, DH1, DS1) and about 50% binds in a different way and/or on a different binding site (Kd1, DH1, DS1). ITC data could be highly valuable for this study and would deserve more attention.

Former Suppl. Fig. 9e → now Suppl. Fig. 4.

We agree with the reviewer that a more thorough, but systematic comparison of SL4 and SL4ext by ITC will add numerical value to our work. We have optimized ITC conditions at room temperature and performed binding studies of the NTD with SL 4 and SL4ext. The ITC-derived affinities are in line with apparent affinities from EMSAs and the observable exchange regime in NMR. We have also included the thermodynamic parameters as obtained from ITC that show enthalpic (favored by the Ext part) vs. entropic (more prominent in the SL4-part) contributions of NTD binding to the two RNAs. The ITC curves show that we do barely reach saturation for the binding to SL4, both in an isolated and SL4ext-context, while the curve is dominated by enthalpic contributions of Ext. This is in line with NMR, where we majorly observe the Ext-type of binding at equimolar stoichiometries of NTD and SL4ext.

We see -as expected- a clear 1:1 binding mode for SL4. We agree that the default two-site binding model does not apply to SL4ext, and in fact it does not reveal fits with derivable parameters. In fact, one would expect to observe an initial 1:1 binding to the Ext part followed by a second, independent binding to the SL4 part. As a result, the binding curve will be an overlay of both and ultimately lead to an excess of 1:1 complexes with Ext, a small fraction of complexes with SL4 and a yet minor fraction of 2:1 complexes (see also NMR data with labelled SL4ext RNA, response to third but last comment). Of note, the medium μM -affinities of SL4 and as the two binding events are not completely distinct we assume a limitation in resolution of ITC for fitting. This gets obvious from the non-intuitive molar ratios of ~ 0.5 and 1.2 , likely caused by the sum up of binding curves, however, highly reproducible. We additionally used the SedPhat tool, which is now stated in the methods section, and a sequential binding mode with two parallel titrations used for fitting. We have replaced ITC data to derive the requested thermodynamic parameters and included replicates as a graphical plot in the new Suppl. Fig. 4.

We have added a statement to the ITC figure caption where we show we are aware of this limitation. *“Note that the sequential interaction of NTD with the two potential, distinct SL4ext binding sites likely leads to a sum up of binding events measured in ITC, and subsequently a mixture of NTD-Ext > NTD-SL4 > 2xNTD-SL4ext complexes. As a consequence, fittable molar ratios are affected while we neglect the low presence of 2:1 complexes (in line with NMR data). See also Methods section for details.”*

We have also added the information of temperature during EMSA runs (for possible comparison) to the methods section.

The authors did not give any details regarding titrations of RNA by addition of increasing amount of NTD. This information is missing in the Materials and Methods section. The dilution of the sample due to the addition of the protein is not indicated. This is especially important for experiments depicted in Fig. 7b and 7c. Page 13, lines 412-413, the authors discussed a “stepwise decrease of imino signals” and concluded that this suggests that the NTD interferes with the equilibrium of the Ext conformation. The decrease of imino proton signal is very weak (Fig.7b) and without other elements and details about the way the titration was performed, this conclusion cannot be stated.

Thanks for making us aware of the missing information in the Materials and Methods section. We included a pipetting scheme and correlating number of scans for those titrations with dilution effects in the Source Data “NMR titration specifics” and made a reference to it in Material and Methods –NMR – section.

Note that for SL4ext we have now included the RNA-observed titration with NTD by means of ^{15}N -labelled RNA, where we confirm the observation of a modulated conformational equilibrium of the Ext part, seen by the selective loss of signals for Ext visible at 278K (new Suppl. Fig. 16,

see also response to comment of reviewer 1 below). This is thus in line with our loss of signals in Ext alone upon titration with NTD and as observed on the 1D-level.

In Fig. 7c, and page 13, the authors described “line-broadening of the SL2 imino peaks”. However, analysis of the titration clearly shows appearance of new peaks, disappearances, shifts and broadening of resonances. This has to be described. It is not only a line-broadening phenomenon.

We agree that our statement on line-broadening for SL2 peaks is not precise enough. We have now rephrased the description as to that SL2 resonances are differentially affected by binding of NTD. As we assume binding takes place at the flexible, invisible SL3 part, we also suggest that this may lead to more than one conformation of, or transient interaction with SL2, e.g. supported by peak splitting of U49. Still we interpret the change of spectra during titration as majorly characterized by line-broadening, which may be caused by an increase in MW, but also the suggested transient interaction with the NTD via the SL2 SL backbone surface (see scheme right to spectra in figure). Importantly, the latter is facilitated by the short/absent linker between SL2 and SL3. The new text passage at the end of page 12 now says:

“Binding to the unfolded SL3 part leads to a stable complex and a reasonable increase in molecular weight of the RNP, which is reflected in significant line-broadening of the SL2 imino peaks (Fig. 7c). In addition, we suggest that, analogously to our data on SL4ext (Supplementary Fig. 16), charge-driven, (in)direct interactions with NTD lead to shifting and new SL2 peaks. Here, the effect clearly underlines the particular interdependence of SL2 and 3, also expressed by their seamless transition from one element into the other (Fig. 6c) and supporting the existence of SL2+3 as a regulatory hub.”

In the Supplementary Fig. 9c, spectra of SL4 and SL4ext are shown at 278K, 298K and 310K. In the Supplementary Fig. 9d, a minor CS differences is shown but the temperature is not indicated in the legend (should be 298K I guess?). Below 298K, spectra exhibit some significant differences and this should be discussed. At 298K, the authors mentioned that the Ext region does not interfere with the folding the SL4 domain but, below this temperature, this should be verified, especially since the titration for Ext was performed at 15°C. The imino protons of SL4ext should be assigned at low temperature by analysis of NOESY and 1H-15N HSQC.

Former Suppl. Fig. 9c/d → now rearranged in Fig. 6 and Suppl. Fig. 11

We thank the reviewer for suggesting to look at CS of SL4ext in more detail. We assigned SL4ext imino signals both at 298K and at 278K. Only at 278K additional peaks for Ext are assignable - as for isolated Ext. We removed former Suppl. Fig. 9c-d, which was inadequately suited to compare SL4 and SL4ext at room temperature. Instead, we now compare the 1D spectra of SL4 and SL4ext at 298K in main Text Figure 6. They reveal CS differences between them locate to the artificial GC base-pair in SL4 construct and natural base-paired U85 and G127 in SL4ext construct. For the SL4 moiety in SL4ext we further observe some spectral changes in dependence of temperatures as now depicted in Main Text Figure 6 and Suppl. Fig. 11, that indicate the folded Ext SL at 278K has an impact on lower stem base-pairs of SL4. The data now directly prove the existence of Ext within the SL4ext construct, and this is best visualized by ¹H-¹⁵N correlation spectra of the RNA. For completeness, a comparison of SL4-only spectra at 278K and 298K is provided in Suppl. Fig. 16.

2D spectra have also been used to monitor peaks upon addition of NTD, both at 278K and 298K. Those now more clearly support the preferred binding of NTD to Ext over SL4. All new data underlying the assignment of SL4ext are now summarized in the new Suppl. Fig. 11.

Note that, to reduce data amount, we removed spectra of RNAs at 310K in the new figures as they do not contribute additional information or value for data interpretation.

In the Supplementary Fig.12, assignments are missing: G85, U89, U120, U115 and G102 should be added. The temperature should be added in the legend. It is surprising that the resonance of g-1 appears with a high intensity. The imino proton of the first base pair in RNA is often broadened due to fast exchange with the solvent. Could authors comment on this? It should also be explained why only imino proton adjacent to bulge or loop undergo line-broadening without altering the rest of the structure, in particular the adjacent base-pairs? Finally, new peaks are showing up at 10.7 ppm: do they arise from the protein or the RNA?

Former Suppl. Fig. 12 → now Suppl. Fig. 16b.

We have put more information to the legend of this titration figure. This also includes explaining the peaks around 10.7ppm, which indeed belong to NTD (see also Fig. 7b for a clearer picture in the titration to Ext RNA).

Note that g-1 in fact represents G85. In SL4, two additional GCs ('g' and 'c') have been added to the natural sequence starting at G86, replacing the wildtype C84-G127 and U85-A126. As such, the first imino proton of base-paired g-2 is barely visible (and was not labelled, which we have changed now), but indeed exchange broadened in our buffer/temperature conditions, while g-1 (G85) is the second base-pair. Note that the SL4 assignment is taken from previous publications (Wacker et al., NAR 2020; Vögele et al., Biomol. NMR Assign. 2021). In our titration, we suppose that the NTD does not mentionably interact with base-paired nucleotides, but temporarily grabs the bulged parts, which is then maximum sensed by the directly neighboring base-pairs, and less or not in more distant base pairs. Also, one needs to consider that we interpret local binding by broadened peaks (seeing the increase in MW), rather than CSPs. Thus, not all effects may be appropriately quantifiable from the titration. We have now adjusted this plot (new Suppl. Fig. 16b) and restrict ourselves to the peaks that are unambiguously traceable upon addition of NTD and significantly line-broaden with respect to the rest of the 1D spectrum.

We, however, agree that a simple 1D-based analysis of an RNA of this size might complicate or falsify interpretation based on a combined shift and line-broadening of peaks (plus the appearance of additional protein peaks). As described above, we have thus now looked at SL4 resonances at RT in the context of the SL4ext RNA (comparable with ITC data) using ¹⁵N-correlation spectra, i.e. the analogous experiment on the 2D-level (Suppl. Fig. 16a). With this, we now provide an unambiguous identification of residues affected by binding of the NTD. We have now described our findings in the second but last paragraph of the results section.

The conclusion and discussion describing the interaction between RNA and the NTD protein suffer from an important missing experiment: the titration of SL4ext with increasing concentration of NTD, as shown in the case of SL4 in Fig. S12. This titration should be performed at 15°C by monitoring imino proton region using one-dimensional and 1H-15N HSQC experiments. The SL4ext should be 15N labeled and assigned as described in the Supplementary Fig. 12 in the case of SL4.

Additional data can be found in main text Fig. 6b, Suppl. Fig. 11 and Suppl. Fig. 16b.

As described above, we have performed those experiments including an assignment of SL4ext.

In the Fig. 1, EMSA experiments are depicted and the NTD shows a stronger affinity for SL4ext in comparison to Ext and SL4. In the Fig.2, the CSP analysis of NTD binding to RNA shows that the SL4ext exhibits the highest values. All these experiments suggest a cooperativity

between Ext and SL4 domains for binding to the NTD. As imino protons are visible at low temperature, the titration of SL4ext should bring important details about the interaction and the respective roles of SL4 and Ext domains.

Former Fig. 2 → now Suppl. Fig. 2b-c.

We agree that the possibility of cooperativity between SL4 and Ext needs to be discussed. We do note that CSPs in Fig.2 (now in Suppl. Fig. 2b) indicate a larger degree of total CSPs for SL4ext vs. Ext. For the EMSAs and using the apparent K_D values we would prefer to carefully evaluate the given numbers. In fact, in ITC we do not see an influence of the presence of Ext for the affinity to SL4, while the Ext K_D is revealed as 10uM (at RT, no ITC was possible at 278K). We think that those numbers are within the fluctuations between methods, and there is no measurable cooperativity between the two sites. As the reviewer suggests, we have also used NMR of ^{15}N -labelled SL4ext at 278K in titration with increasing amounts of NTD, which shows that Ext peaks are first line-broadened beyond detection. However, the absolute increase in MW up to almost 60 kDa for the subsequent 2:1 complex, leads to a significant increase in line width at low temperature, which is also evident in some, but not all peaks of SL4. In turn, the same titration at 298K shows that only residues close to the transition to the Ext part are affected at low ratios of NTD. We thus conclude that cooperativity is no major driver of additional affinity of SL4ext vs. Ext. Nevertheless, we clearly support that avidity, or the possibility of an encounter complex caused by the presence of charge in SL4 may additionally drive complex formation with specific nucleotides in Ext.

Reference #39 should be updated (now published in a journal).

Reference #39 (now #34) has been updated.

In conclusion, although the work is original in the field, several aspects should be seriously improved. Its significance could be strongly improved by obtaining a structure for the stable SL4ext-NTD complex.

We definitely agree that structures of the NTD with one or more of the herein suggested RNAs would greatly aid in understanding atomic details of target selectivity. We have undertaken strong efforts for co-crystallization with Ext (including soaking and crystallization at 37°C), but currently end up with the apoNTD or insufficient scattering. We assume this is majorly caused by the lability of the RNA, which apparently is an important feature for recognition by the NTD, but at the same time prohibits crystallization. The need of low temperature for Ext to be studied as a more homogenous RNA via NMR likewise complicates the determination of a complex structure.

Reviewer #2 (Remarks to the Author):

The article by Korn and colleagues is a detailed investigation of the RNA-binding properties of the N-terminal region of the N protein from SARS-CoV-2, using ligands derived from its genomic RNA. The N protein contains both specific and non-specific RNA-binding properties, and yet still functions in a precise manner during the viral cycle. As such, its RNA recognition properties are complex. The authors have done a good job to provide insight into what may define the specific elements of the interactions despite an apparent high degree of variation in the known N protein binding hot spots in the genomic RNA. I find that the series of experiments are performed well, and encompass a wide range of biophysical approaches to study the topic. This study is an important contribution to the understanding of the N protein function. I have no major concerns regarding the work, but nonetheless I have some questions to be

addressed:

It would be great to include a supplementary graphical figure to go with the ligands in Table 1. It takes a bit of effort to determine exactly where the constructs start and stop within the genomic RNA structural elements. This also relates to the next point.

We have now included such an overview, which lists all RNAs with their approximate position in the SARS-CoV-2 genome, their numbering and their detailed secondary structure in the new Suppl. Fig. 1 (see also comment to reviewer 1). We have also updated Table 1 and Supplementary Table 1 and added helpful comments to the underlying sequences.

Given that N protein has been linked to binding ssRNA in the past (such as on line 127) would it not have made sense at some point to extend one or more ligands to include the ssRNA parts before or after the stemloop(s)? This is partly addressed in the final experiments (such as Suppl. Fig. 10b,c) but could be more explicitly referenced in the manuscript. I am not suggesting a requirement for new experiments (although perhaps some additional results have already been obtained that addresses this question?). However, I think it should be more clearly stated that the N protein RNA ligands did not include the adjacent ssRNA (even though they may actually be recognized by protein N..)

We found this comment particularly inspiring, and thus addressed it with additional experimental input, summarized in Suppl. Fig. 14. It displays the interaction of the NTD with additional RNA elements of this type. The 5' UTR contains one further stemloop, SL1, that is followed by a – predictably - unstructured stretch of 11nt, termed here ext^{SL1} . We used $SL1_ext^{SL1}$ in homology to $SL4ext$ (= $SL4_ext^{SL4}$) and found that it is an SL4-like binder and, as such, less preferred. Interestingly, swapping the extensions of SL1 and SL4 showed preferred binding to $SL1_ext^{SL4}$ over $SL4_ext^{SL1}$. This indicated that indeed the “Ext” following SL4 is specifically bound, irrespective of the neighboring stem-loop's nature.

In addition, and summarized in Suppl. Fig. 13, we looked at a 1-kb-stretch in the coding sequence that had been described by the Doudna lab (Syed et al., Science 2021), termed P9, to be of relevance for N-mediated genome packaging (see also our response to reviewer 4). Within P9 we found a particular stemloop in combination with an expectedly labile structured extension, which we termed P3. The NTD bound to the stable stemloop (P3_A) in an SL4-like mode, while the extension (P3_B) and the combination of both (P3) are SL4ext-like binders. This supports our hypothesis on NTD binding preferences and underlines the utilizability of the signature peaks as a quick and easy readout for preferred targets.

In the very first results paragraph we have now clearly indicated, that our work focuses on defined elements in order to gain information on parameters that drive specificity for the NTD and does not cover a comprehensive analysis of the influence of flanking regions. Beyond that, we are aware of the option that in general regions before or after the isolated targets (be it stem-loops or other elements) can add to recognition by N, also via the other domains, which is part of our discussion. Note though, that based on technical reasons of RNA in vitro transcription, folding homogeneity and stability, our SL constructs usually do contain some kind of overhangs as is shown in the new Suppl. Fig. 1.

I found the careful study of generic ssRNA binding vs specific ssRNA to be particularly useful to understanding the N protein RNA preference. Just out of interest for potential discussion, do you think that in the cell the transient dsRNA-ssRNA segments might also be important hot spots, in that since they are not completely ssRNA, they might be 'protected' from interaction with the plethora of ssRNA binding proteins in the cell? That is, the partial dsRNA nature would prevent or lessen recognition by many of the host RNA-binding proteins? Therefore, N protein could still be specific for ssRNA elements, which would be clear in vitro, but the weak dsRNA/ssRNA character could be critical within the cell.

We thank the reviewer for this interesting and well justified perspective. We indeed believe that the suggested mode could have a contribution to specificity towards viral RNA over host despite its obvious ssRNA-preference. We have included this option in the discussion (see lines 436-440) and also indicate it in the new final summary panel/model (see next comment).

A final model would help gather the main findings together. I find that the current final model (Suppl. Figure 13) does not provide much help in summarizing the main results and is perhaps mostly speculative. I would instead suggest to provide a final summary that focuses on the favored targets of protein N (weak dsRNA that have a high ssRNA population) and any other main aspects of the interaction and mechanism that are derived from the results.

We agree that our previous model in Suppl. Fig. 13 was somewhat speculative and that a focus on the main outcomes of this study will be more desirable. Thus, we provide a new model in the new main text Fig. 8, highlighting the RNA preferences and their labile fold/ss nature, in combination with the NTD signature peaks.

In accordance with the exchange of the model figure we shortened the discussion section (see first comment of reviewer 1).

In the first two paragraphs of the results, there is some discrepancy between the statements that the NTD differentiates by size (line 111) and that the Ext and SL2+3 have similar affinities (line 124). This would also apply to the large, but poor binding SL4. Maybe the size dependent statement can be removed?

Indeed, this was not written clear enough. We shortened this part and rephrased these somewhat inconsistent statements; summarized in the very first results paragraph.

We suggest to keep the potential role of large, charged surfaces as given in the branched SL5 element for unspecific affinity to NTD as part of the text, as to justify why we do not focus on SL5 elements here any further.

Figure 2 would benefit from a secondary structure diagram above the graphs, just as in Figure 3, to help the reader identify where the CSP regions are in relation to the protein.

We have added the diagram in former Fig. 2, which is now Suppl. Fig. 2. For convenience, we decided to add the diagrams to all NMR data plotted over NTD sequence where we think this adds a helpful context.

In the paragraph from lines 182-187 the importance of the multiple field strength is not clear, nor is the conclusion that the b-hairpin displays a rigid-body dynamic. Perhaps this paragraph can be removed since the key finding is already mostly part of the last sentence in the previous paragraph (lines 179-181).

The importance of multiple fields has been included for comparison with an earlier study of NTD from SARS-CoV (Clarkson et al., JBNMR 2009). Similar as to what is suggested there, we see an important role for the coupled dynamics of the β -hairpin and the N-loop for a concerted function (see design of an appropriate mutant as response to reviewer 4). We thus suggest to leave this information included in our data, but agree that the statement on rigid body movement is not fully supported by our findings, but somewhat speculation. We have now rewritten this part and shortened it, such that no redundant information is any further given (lines 144-151).

Interpretation of the dynamics results in the section from 296-328 requires additional clarification or instead a clearer explanation of potential relevance of the observed dynamics

and how it relates to RNA binding. For example, what would be an advantage to coupling the observed dynamics to RNA preference.

To clarify the role of NTD dynamics in RNA-binding, we used a mutation in the hairpin hinge region (S105I) to abolish a polar interaction of this residue with the adjacent N-loop residue Q58, as depicted in the new main text Fig. 3 and new Suppl. Fig. 7 (see also the response to reviewer 4, in context of additional NTD mutants). This NTD mutant shows a loss of significant reporter peak activity (i.e. capability to sense/report on preferred RNA elements), which underlines the crucial role of dynamics for RNA discrimination. The new findings on NTD-internal dynamics as well as the other mutants are now included in the main text with a new, separate section.

Although the experiments are well described for the last two sections of the results, I found several places where the text was unclear. For example, lines 401-409 are not clear to mean regarding the main point that is being made - perhaps some words were accidentally removed? Or as another example on line 436, maybe the end of the sentence should be 'with an increase in temperature' or something similar.

The last two parts of the results section have been comprehensively rewritten and new data added. With respect to the mentioned sites, we rephrased the section and hope our main point becomes clearer for former lines 401-409, which now says: *"In contrast, we tested another RNA that we expected to categorize as preferred binder. The P2 RNA element is located in the ORF1a/b region between SL12 and 13 (Supplementary Fig. 1 and 12b). Consistently, both the Ext region between SL4/SL5 and P2 had been described as hotspots for cross-linking with N and are expected to specifically trigger LLPS in the context of genome packaging. As shown in Supplementary Fig. 12c, P2-binding to NTD in the SL4ext-mode unambiguously supports the inherent robustness and validity of our combinatorial approach."* Note the shift in figure panels from Suppl. Fig. 10 to 12 and the swap of panels b and c for clarity.

Similarly, for the former line 436 the sentence now reads: *"Comparative aSEC at 5°C vs. RT revealed significantly increased RNP amounts for SL2+3 and Ext, and to a lower extent also SL4ext (Supplementary Fig. 15d and e) with an increase in temperature."* Also here, note the shift of the underlying panels from main text Fig. 8 to Suppl. Fig. 15.

The figure icons for Fig. 8c,d,e are not the easiest to follow. I assume that the number of RNA that are stemloops or curved lines represents a shifting population? However, at present it looks a bit like the RNA is oligomerizing on the protein or some other more complex behavior. Unfortunately, I do not have a specific recommendation for increasing clarity, except perhaps in figure c and d to indicate that protein N NTD shifts the equilibrium towards the ssRNA, maybe by placing equilibrium arrows between a single unfolded and folded RNA, and have protein N beside the arrow from stemloop to ssRNA. And then increase the arrow width at different temperatures to reflect changes in the population. But this is just one possibility.

Indeed, the icons could be misinterpreted, as they were. We provide a new representation, based on the reviewer's suggestion and hope to delineate this context clearer now. Note that the previous Fig. 8 has been shifted to Suppl. Fig. 15.

For the Discussion, my other suggestion is to remove the Supplementary Fig. 13 and perhaps even the paragraph from 537-547 since it is not that clear in its current form, and is outside of the main theme of the manuscript.

We shortened the discussion significantly, including this section, and -as detailed above-removed the former model by a less speculative one (new main text Fig. 8).

Tiny point: some figure panel letters are missing in the text (such as line 229 should be Suppl. Fig. S6'A'), and in line 199 it is not clear which figure the 'panel a' is referring to.

We thank the reviewer for pointing us at the missing panel letters. We have added all letters after rearranging figures and hope they are complete now.

Reviewer #3 (Remarks to the Author):

This is a potentially very interesting story describing the biophysical characterization of an important protein-nucleic acid complex and there is a good degree of experimental design (lots of different constructs, lots of tests).

However, much more information needs to be provided about the SAXS data before this could even be considered for publication. The authors do not show raw SAXS data in the text, only the result of processing which can (but does not have to) be impacted by parameters selected during the analysis. The lone exception is found in Supplementary Figure 6.

We absolutely agree and now provide raw data, equivalent to the free NTD, for all RNAs and the complexes both in the new Suppl. Fig. 8 and the SASBDB (see response below for more information).

It is stated in the text that “the quality of our SAXS data is well reflected in an excellent fit of an experimental NTD scattering curve compared with back-calculated SAXS scattering from an available crystal structure (Suppl. Fig 6).” To my eye, Figure 6 shows a (partially) aggregated sample and a rather poor fit at the most important, low q values, which assess the heterogeneity of the sample. The authors do not appear to recognize that this low q signal is a problem, thus raising concerns about the interpretation of all of the rest of their SAXS data. If the NTD is not monomeric, how can association be properly characterized? The following statement in Methods was also concerning: “initial points [were] removed to enable shape fitting”. Does this imply that aggregation was evident?

We agree that the previously included raw scattering curve indicates the potential presence of higher oligomers for the NTD and the fit appears less good at low q values, while all downstream derivable parameters indicate a clear monomer (D_{max} , R_g , MW when comparing experimental and theoretical values, see Table 2). For the NTD we carried out additional experiments and confirmed its pure monomeric state, even at very high concentrations, by analytical SEC versus a MW standard as well as by another round of SEC-SAXS coupled to SEC-MALLS (Suppl. Fig. 8a-c). Both experiments reveal the absence of any higher oligomers, which is also in line with our NMR-derived MW, taken from the total correlation time calculated from ^{15}N R_1 and R_2 relaxation rates (see Fig. 5d and e). We thus have no doubt about the purely monomeric NTD, which is also supported by previous studies as given in the references (Dinesh et al., 2020; Redzic et al., 2021; Ye et al., 2020).

Of note, all SAXS measurements have been carried out by mail-in and remote data acquisition or as service in a semi-automated set-up due to ongoing restrictions for in-person visits at synchrotrons. We thus do not have full control over sample states after delivery and freeze-thaw cycles, while we do know that the NTD (see above) and is free from oligomers. We likewise find an indisputable major monomeric population of all RNAs in our lab (native gels, SEC profiles, see comment below). That was the reason for us to ignore initial low- q points in individual cases, as they are also being ignored by the automated fits. However, we have contacted the beamline staff as to get an opinion on the presence of small amounts of higher-MW species or contaminations, which also appears present in some of the RNAs (see Suppl. Fig. 8). It turns out that one option is the capillary, which is improperly or not at all rinsed

between fractions in a SEC-SAXS run (for technical reasons). Also, to some extent radiation damage might apply as a consequence of longer sample exposure (less concentrations in SEC-SAXS). We have included the beamline staff response for your information here:

“Obviously, you have not removed any points at low-q yet as all curves seem to have the same amount of points. Indeed, the autosubtracted curves look like aggregates or maybe radiation damage. While I agree that aggregates should be removed in SEC (depending on the size compared to the protein of interest), radiation damage and / or capillary fouling may easily happen in SEC.

It is very common to investigate the SEC-SAXS output in such a way that varying sample and buffer ranges are used. You seem to have done this and the resulting Manualshort looks absolutely acceptable. I have overlaid it with the other curves and except for low-q, they look almost identical.”

Indeed, for the NTD we do find less pronounced oligomers based on upturning low-q when reducing the number of frames used from the SEC-SAXS profile's peak. However, this also leads a reduced S/N and ultimately does not any longer cover the full width of the protein peak and is thus considered an inappropriate bias in data analysis. Further, it does not affect Dmax, Rg and MW clearly indicating that the major species is a monomeric protein.

We have reanalyzed all SAXS data with more care and also decided to exclusively include SEC-SAXS runs for a better comparison and to save space as well as to combine Tables 2 and 3 into one. No points are artificially removed. All SAXS data processing is now more comprehensively described in the Method section, data are deposited in the SASBDB, and the new Table 2 shows the comparison of expected and experimental Dmax, Rg, and MW for all species (based on models for RNAs), while all Crysol fits are given together with the raw curves in the Suppl. Fig. 8.

To not ignore the presence of potential high-MW contaminants in the raw data curves we have included a statement in the legend of the respective figure.

Additional concerns arise from the statement that “SEC-SAXS allows analyzing individual fractions of NTD, RNA and RNPs”, and other references to the use of SEC-SAXS to measure specific fractions. If Figure 6 is the result of a SEC-SAXS experiment, the columns may not have properly separated the fractions. As a result, it is hard to accept the interpretations of the SAXS data, without further proof that the samples are in the monomeric state.

We assume this reviewer refers to the previous Suppl. Fig. 6 (now Suppl. Fig. 8). We agree that for individual runs the separation of RNPs from free RNA and/or protein is not fully guaranteed as we are limited by the columns (i.e. also time and amount of samples). The choice of columns is thus based on compatibility with the beamline setup and the possibilities in our lab. The herein used columns are the optimum compromise considering sample amount (esp. expensive RNAs), resolution, beam time and control over contaminations. Of note, we have used the same type of columns for in-house SEC runs and SEC-SAXS and SEC-MALLS. We do however not question the monomeric state of the NTD (please see response to point 2 above). For some RNAs, small contributions of oligomers from in-vitro transcribed stocks are evident in apo-RNA SEC(-SAXS) runs. This is exactly why SEC is used in combination with SAXS and MALLS as to allow looking at components within mixtures and especially in order to derive clean SAXS parameters. Certainly, larger/longer columns might push resolution, but the increased need in input material by a manifold factor and incompatibility with SEC-SAXS do currently restrict us to the columns we have used here.

Further, note that the SEC profiles shown in the Suppl. Figure are plotting absolute scattering intensity over retention volume. Here, based on its larger scattering propensities, the signal is largely dominated by RNA. Also, the new Table 2 now shows the SEC-SAXS-derived

geometric parameters of NTD-RNA complexes, which are in very good agreement with theoretical values (as far as one can say for the MW, no experimental or model complex structures do exist). We have also adjusted the RNP frames to be picked from SEC profiles to avoid unintentional mix-up with free components (see Source Data).

We thus are convinced that samples as they are used for SAXS-based analysis are sufficiently separated on our columns and the fractions of interest do primarily represent monomers or 1:1 complexes.

The authors should read and follow the guidelines put forth in: “2017 publication guidelines for structural modelling of small-angle scattering data from biomolecules in solution: an update”, *Acta. Cryst.*, D73, 710-728 (2017). Please use the table in that document to provide information about model independent parameters to allow a fair assessment of the SAXS data. And please show the raw data.

To comply with state of the art standards in SAXS sample quality and SAXS data acquisition, processing and exploitation we have now deposited all underlying SEC-SAXS data in the SASBDB, where raw data by default undergo quality control and in our case support the good quality of SAXS data in our manuscript. This is now stated in the text, and accession codes will be made available. For the review process we have added links to the deposited data, which are under final revision in the SASBDB. The depositions include raw scattering curves as well as fits to the structures/models. We have however included an overlay of raw scattering with a Crysol-derived theoretical scattering from models for convenience (Suppl. Fig. 8).

In addition, we thank the reviewer for pointing us at the mentioned publication, from which we derived and included an additional table (placed in the newly submitted Source Data), that provides a fair assessment of our SAXS data by additional measures. We highly appreciate this quality-oriented initiative and hope to convince the reviewer of the trustworthiness of SAXS data shown in the manuscript.

Other comments:

I disagree that the changes in Figure 4a shown for SL2+3 are any more (or less) significant than those shown for SL4. The D_{max} is a ‘selected’ parameter, and given the above raised concerns about aggregation, it needs to be double checked. Please show the raw profiles, the answers will be obvious from looking at the curves.

We apologize if we have not been able to more precisely point out what exactly this seemingly inconsistent comparison might mean. In fact, we had discussed in the text the possible explanation that the binding of NTD to SL2+3 opens up the (transient) SL3 stem-loop, which might on average lead to a smaller D_{max} , compensated by the tightly bound NTD. This is less if not at all not likely for SL4 seeing the NMR data. We would like to underline that Fig. 4a is not exclusively chosen to provide D_{max} as the readout, but the entire $p(r)$.

We fully agree that solely looking at $p(r)$ curves and D_{max} is insufficient for monitoring binding, at least in this setting. This is exactly why we additionally looked at the change in Porod volumes in Fig. 4b.

We have rephrased the respective passage in the text (transition from page 7 to 8) and hope to appear clearer now for the reader. As mentioned above, we provide all raw profiles with the manuscript now.

The computed MW for Ext is inconsistent with the quoted R_g , and the asterisked comment that might explain the difference does not appear in Table 2. In fact, the R_g suggests that the sample is aggregated, in conflict with the MW. What are the error bars?

We thank the reviewer for spotting the misplaced asterisk, which was set unintentionally.

We agree that in the former Table 2 both the D_{max} and the R_g for Ext appeared a bit too large for the MW. In fact, the Ext theoretical MW value is supported by SAXS experimentally by means of the derived Porod volume/invariant approach (yielding 7.1 kDa). They also appeared too large relative to the provided model, although we would like to underline that all theoretical R_g and D_{max} values for RNAs are derived from models, not experimental structures. As stated above, we have now, for better comparison, exclusively used SEC-SAXS data, i.e. also for free RNAs. This significantly reduces the calculated D_{max} (4.34 nm) for Ext, and is in line with the theoretical value. The R_g is now given as 1.54 nm, and the raw curve does not show indications of aggregates. A possible reason for this deviation might however be that the real Ext structure is still not covered well enough by our model, e.g. seeing its intrinsic flexibility.

The MW derived from SAXS for Ext is now given as 9.5 kDa, in accordance with the value suggested by the SASBDB, which is calculated from Bayesian inference and appears too high. We have added an asterisk to the Ext table entry which allows us to state that a Porod volume derived value for the MW of Ext perfectly fits the theoretical 7.1 kDa (so is also the one derived from the 'size&shape' option). We will however for congruence of all RNAs and NTD stick to the values obtained from Bayesian inference in the table, and the values are in good agreement for all other species tested here.

We have now also included all errors as derived from ATSAS for R_g values and a total quality estimate for the $p(r)$ curve in the SAXS-related Source Data File (see response to comment above).

The sample concentrations for the SAXS data were never reported. Concentration is critically important for SAXS experiments.

We are very grateful for pointing at this lack of information and absolutely agree. We have added all concentrations and injection volumes of SEC-SAXS samples in the Method section or the overview table in the new Source Data. Please note that apparent concentrations in SEC-SAXS fractions used for the acquisition of scattering frames (i.e. main peaks) somewhat remain difficult to estimate and are subject to a case-by-case basis. We have given an estimate of this for free RNAs and the NTD together with the values for injection in the Source Data file to relate the SAXS-derived parameters to a realistic concentration.

The authors, in several places, comment on the 'excellent' work performed by others. This seems unusual.

We have removed the attribute 'excellent' in those contexts and confess it is rather not suited for such a manuscript.

Reviewer #4 (Remarks to the Author):

Korn et al describe "the preference signature of SARS-Cov2 nucleocapsid NDT for its 5' genomic elements". I find the manuscript difficult to read and I don't see the bigger picture/general interest for this work. If we want to understand the binding preference of nucleocapsid NDT than learning how it binds a handful of structures is not sufficient.

We have worked over the manuscript, especially addressing its length and redundancy, to make it more readable. We have also included new data to emphasize the relevance of this study and our findings in more breadth, which are detailed below (see also responses to the

other three reviewers, as well as the attached document “Summary of changes” for a comprehensive overview).

Do the authors gain any predictive power? Can they predict whether (or which) the NTD would bind to 5'UTR structures of other coronaviruses? What about other families of RNA viruses? How does the NTD domain discriminate between RNA stem-loops from coronavirus verses stem loops within cellular mRNAs?

Our study was driven by the question what are the specific elements to be recognized by the N NTD in the SARS-CoV-2 genome, starting from the highly regulatory 5'UTR/genomic end. We do not rule out that SARS-CoV-2 N NTD would in general also recognize suitable elements in other RNA viruses, especially seeing the high degree of conservation both in N and the RNA within coronaviruses. To date, there seems little known about the need of a virus' nucleocapsid to specifically distinguish viral RNAs in case of co-infection though. But, we agree that a major challenge is given by the presence of host (m)RNA to compete with viral elements, considering the general ability of N to bind to host mRNAs' 3'UTRs (Nabeel-Shah et al., *iScience* 2022). We would, however, like to stress out that current literature suggests a model in which SARS-CoV-2 RNA makes up more than two thirds of all mRNA soon after infection, i.e. prior to packaging (see e.g. Finkel et al., *Nature* 2021), moreover in the relevant cytoplasm (Zhang et al., *Sci Adv.* 2021). That, in combination with a distinct cellular locus (the DMVs) of packaging for viral RNA -possibly connected to active recruitment of N by M (e.g. Lu et al., *Nat. Comm.* 2021) and Nsp3 (Bessa et al., *Sci Adv.* 2022; Wolff et al., *Science* 2020)- will lower the likelihood of engaging with an excess of host RNA. Nonetheless, specific and stoichiometric recognition of own RNA remains a challenge, and we do not rule out that consensus motifs of the NTD are present in host RNAs. Potentially, it is the particular arrangement of target elements in the viral RNA to be recognized by N oligomers and including its IDRs (see a recent publication from Pontoriero et al., 2022, and our former model in the previous Suppl. Fig. 13 for this) which altogether lead to a sum-up of the slight NTD preferences. It is important to keep in mind that the NTD affinity for RNAs per se is relatively low, but subtle differences will emerge in larger RNPs, i.e. stoichiometries and N oligomers. The complexity of N makes us assume this would be beyond the scope of this story, while we certainly discuss this mechanism in our manuscript in order to embed our findings. We would like to re-emphasize that our approach puts a major technical focus on the unambiguous NMR-driven readout of relevant NTD-bound RNA motifs and the related robustness of RNPs in simple aSEC experiments and sheds light on the potential to identify such motifs from the enormous pool of arbitrarily bound RNA elements. This is now more focusedly included with the new summary model in Fig. 8 (see response to reviewer 2)

Further, and in line with a comment by reviewer 2 (see above), we favor the idea that the recognition of preferred viral elements may be facilitated by the ambiguous nature of ssRNA that temporarily and dynamically appears in secondary structures like SL3 or Ext. This particular feature might be more pronounced in viral RNA as compared to host RNA and thus represent a driver of specificity. We have now included this in the discussion.

However, we absolutely agree that our findings in general need to be judged by their potential to derive a “consensus motif” or motif characteristics utilizable for suggesting and validating potential binding sites of the NTD within the SARS-CoV-2 genome, which should show identical characteristic behaviors in NMR.

To this end, we had already included the P2 element in our initial submission that had been suggested as a target of N outside the 5'-UTR, and for which no particular involvement of the NTD had been suggested before. As outlined in the new Suppl. Fig. 12, this element fulfills all criteria and experimental expectations with regard to the CSP patterns of a preferred binder.

We now added another additional RNA element that we expected to classify as a preferred target. We identified a suitable element within a SARS-CoV-2 genome stretch, which represents the SL4-Ext setup. The 1-kb genomic context was recently suggested to significantly promote RNA packaging by the Doudna lab (Syed et al., Science 2021). We defined an element, here termed P3a/b, for which we confirm the anticipated binding behavior with the NTD including a low-affine SL and a high-affine target site adjacent to it (using NMR CSPs and aSEC in Suppl. Fig. 13). Of note, this particular region has previously not been confined as precise with respect to the actual packaging signal.

The main claim of the abstract “We show that the domain uses a set of flexible sensory residues to read the intrinsic signature of preferred RNA elements for selective and stable complex formation within the large pool of available motif” is not supported by structure/functional analysis – the authors should mutate some of this “sensory” residues and predict the outcome of these mutations in terms of RNA structure recognition and binding.

Indeed, we do agree that NTD mutations will help to corroborate our claims and are grateful for this suggestion. Guided by available high-resolution structures of the NTD we decided to address three different features:

1) As a general control we used an earlier suggested mutant (R107A) to show that binding is impaired or abolished for the 5geRNA elements.

2) As suggested by the reviewer, we then used mutations of the residues G60 and G99, representing direct sensors, to show that the other sensors are still functional, respectively. That means, the sensor residues themselves do not steer the NTD’s capability of discriminating between RNAs, but merely sense them. This observation strongly underlines the potency of our readout for NTD RNA selectivity to be sensed by NMR.

3) Finally, to address the role of dynamics in the NTD for discriminating RNA elements we chose to mutate residue Ser 105, whose sidechain acts in linking the basic finger with the N-loop. The exchange of Ser105 by Ile indeed led to disruption of this interaction as shown by impaired hetNOE data. Strikingly, we found this mutant to be incapable of discriminating SL4 and Ext with respect to their characteristic CSP patterns. This indicates a strong role for the level of flexibility necessary for the NTD to identify RNAs specifically. See also response to reviewer 2 for this mutant.

The rationale for mutant design has been included in a new main text figure (Fig. 3, supported by Suppl. Fig. 7). We hope to convince this reviewer with the new data with respect to his suggestion.

Overall, the authors perform a lot of detailed biophysical characterization of the NDT but lack testable predictions and are somewhat repetitive such as some of the figures seem better suited for the supplementary. For example, it wasn’t clear to me is Fig 3c entirely consistent with the gel shifts from figure 1D?

We have -despite the addition of new data as suggested by the reviewers- undertaken substantial shortening of the text. See in particular our response to reviewer 1.

We would prefer to leave Fig. 3C as part of the main text as it exemplifies the running behavior of NTD in complex with target vs. non-target RNAs (here SL4). This simple readout is transferable to other tested RNAs (see all the non-5’geRNAs in Fig. 7 and Suppl. Fig. 1, 6 and 13), and here particularly shown for SL2+3, SL4, Ext, SL4ext to compare it to NMR CSPs. The underlying affinities are in line with the apparent affinities from EMSAs in Fig. 1D. For SL4 vs. SL4ext, see also our new ITC data and response to reviewer 1.

Each figure should emphasize the critical new information that is being added. In my opinion this is careful and rigorous work but without mutational characterization or deriving general

principles for discrimination of structure motifs, this work is better suited for a more specific journal.

To communicate our findings and interpretations in a more focused and comprehensive way we have put more effort to the particular emphases of each paragraph and its figure(s), incl. modified headlines, figure headers, rearrangement of in and between the main text and supplement. Together with the shortening of the manuscript we hope to achieve more compactness and overall readability to stress out the novelties. Having included additional data, which we believe are of large value, we hope to convince the reviewer of the suitability for this journal.

REVIEWER COMMENTS

Reviewer #1 (Remarks to the Author):

The authors have addressed almost all of my concerns and they did an overall good job. I recommend publication of the manuscript provided two points are corrected:

(1) in the supplementary Figure 11, panel a, U145 is indicated twice. The assignment “U133/U145” has to be corrected by “U133/U144”, as indicated in Figure 6b.

(2) I still have serious problems with the ITC data. First of all, in their response to reviewer’s comments, authors discuss their optimized results obtained by “ITC conditions at room temperature”. Room temperature doesn’t exist for ITC (and doesn't exist at all by the way). In the Supp Fig 4 legend, it is stated that data were fitted with a two-step sequential binding model, but this is only true for the NTD+SL4ext data, not the NTD+SL4 data that were fitted with a single binding site model in the revised version of the manuscript. This should be corrected in the figure legend. The stirring speed of 500 rpm is quite unusual for VP-ITC (usually 310 rpm; higher rate desirable for experiments where aggregates are formed; was this the case? Could authors comment on this?).

In addition to these details, I'm still highly skeptical about ITC data and interpretation. First, there is no mention in the ITC method section of blank experiments and of a negative control experiment with an unrelated RNA of similar size. These experiments seem to me crucial insofar as the experimental ITC data are really very weak (low signal/noise and extremely weak enthalpy) and the obtained K_d for SL4 is more or less within the limits of what can be measured in biomicrocalorimetry: according to the expected pI of the protein (around 9.6), there is necessarily an affinity for any (negatively-charged) RNA which is not negligible due to non-specific electrostatic interactions, especially since the experimental conditions have been modified to be relatively low in salt (50mM KCl only). This should be clarified.

Second, about ITC data for NTD-SL4 interaction, the authors processed these data with a one-site model (I wouldn’t, however, claim – as stated in the response to reviewers - that the expected 1:1 binding mode is clear!), thus excluding any unspecific interactions between the protein and the RNA, which looks quite unexpected according to the high pI for the protein and the low salt conditions.

Third, about the ITC data for NTD-SL4ext interaction, I do not understand the arguments put forward by the authors to explain the model used and the observed stoichiometries (“...as the two binding

events are not completely distinct we assume a limitation in resolution of ITC for fitting. This gets obvious from the non-intuitive molar ratios of ~ 0.5 and 1.2 , likely caused by the sum up of binding curves..."). If a correct model is used for ITC data fitting, stoichiometries derived from the model should be correct, in agreement with the model, and no limitation of resolution limit applies. This can possibly be tested using a reverse titration and a global fit. Again, with this SL4ext RNA fragment it would be surprising to observe no signal from a non-specific interaction given the protein pI and the salt conditions.

Reviewer #2 (Remarks to the Author):

I thank the authors for their careful and complete response to my concerns and suggestions, and have no further comments for the manuscript.

Reviewer #3 (Remarks to the Author):

I appreciate the time and effort that the authors put in in addressing the comments from the last draft. However, interpretation of the SAXS data is still unclear. Here are my questions and concerns.

1. In the last version, there was clear evidence of a problem at low q . The authors reply that this is potentially due to radiation damage, which should be concerning. If the sample is damaged by the beam, how can they trust the data? Usually most experiments begin by finding conditions that avoid radiation damage, by reducing exposure or adding attenuation.

2. It is hard to piece together exactly what was the experimental procedure. Was the aSEC done in the lab, prior to shipping the sample? Hence, days in advance of the beamtime? The SEC-SAXS shows that the sample refractonates on the beamline. How do you know that you picked the right fraction to look at? It seems that the largest fraction was selected, but is there another way to be sure that this is the correct one? Is the frame number reproducible between runs? The data are plotted in a way that makes it very difficult for a reader to assess (and this should be something that the authors report on). Please justify how the proper fractions were determined.

3. In addition, the presentation of curves in Supp Figure 8 is quite confusing. Why are the x axes different for the SAXS profiles? It is quite hard to compare the curves. Furthermore, the points are so light and hard to see, that it is hard to agree with the statement that their fit is 'excellent'. I see significant deviations from the pdb generated curve, and I cannot even resolve the error bars.

4. The radius of gyration values from Table 2 cannot possibly be real.

5. Assuming that the SAXS data are OK (which they might be, but I cannot be confident, given the concerns raised above), the authors MUST provide compelling argument that the Porod volume for a protein RNA complex can be interpreted as they do in the manuscript. When two molecules associate, it is quite reasonable to expect an increase in the Porod volume, so this seems fine; however I am concerned that they are overinterpreting the changes as a remodeling of the RNA. Specifically, they are looking at the interaction of an RNA with a protein. These molecules have different electron densities, so I would have to very carefully consider how these different densities affect the computation of the Porod volume. If there were two proteins associating, the molecular density in the $I(0)$ would cancel out the term in the integral, but it is certainly not obvious to me that this same cancellation occurs for two interacting molecules with different electron densities. If this has been worked out (which it may well have been), the authors need to provide a reference describing the calculation. They might work with a SAXS expert to do the calculation otherwise.

6. Additionally, the data show some artifacts at the low q values, both the increases discussed above, and now some downturns that may be the result of interparticle interference effects. Since the $I(0)$ value is so critical to the computation of Porod volume, it must be properly obtained, and the lack of acknowledgement of this, suggests that the authors might underestimate the difficulty of the analysis they are proposing. This is definitely a non trivial experiment.

7. Please carefully proofread the table of SAXS parameters. There are some errors. The author is Trewhella, not Trewhelly.

Overall, the analysis **might** be correct, but there are many details left unjustified. Given the concerns about the data interpretation, these concerns must be addressed before it could be considered for publication. Because of these concerns, I did not review their conclusions, as more confidence in their methods needs to be provided.

Reviewer #4 (Remarks to the Author):

The authors took a lot of care and thought to address concerns and substantially improved the manuscript. I highly recommend it for publication.

We thank all four reviewers for appreciating our efforts to improve the manuscript. We here hope to sufficiently address the remaining open concerns. In addition, and in accordance with the journal guidelines and policies, we have undertaken the following editorial requests:

Reporting of our SAXS data and methods should now comply with the community guidelines as requested (see our responses to reviewer 3, which includes citation of those guidelines/the underlying reference paper). We have included the SAXS data collection as Supplementary Table 2 and in the Source Data file (see new SAXS experiments), and it is also cited in the main text.

We have marked edits of the second round with blue color in the tracked-changes-versions of manuscript files with respect to the first revised version.

Find below the detailed responses to reviewers 1 and 3.

Reviewer #1:

The authors have addressed almost all of my concerns and they did an overall good job. I recommend publication of the manuscript provided two points are corrected:

(1) in the supplementary Figure 11, panel a, U145 is indicated twice. The assignment “U133/U145” has to be corrected by “U133/U144”, as indicated in Figure 6b.

We thank the reviewer for pointing us at this mistake. We have corrected the assignment both in panel a and c of Suppl. Fig. 11.

(2) I still have serious problems with the ITC data. First of all, in their response to reviewer’s comments, authors discuss their optimized results obtained by “ITC conditions at room temperature”. Room temperature doesn’t exist for ITC (and doesn’t exist at all by the way).

We absolutely agree and have now precisely given ITC temperature settings (298K). We have also replaced “RT” with “298K” at all appropriate sites in the manuscript for NMR acquisitions, where precise temperature control had been performed (inside the spectrometer).

Apart from ITC, we have tried to homogenize and more precisely provide temperature information throughout the manuscript. For experiments where we do not know the precise room temperature we have left the “RT” as such, but define the underlying range in the Methods (*Experimental temperatures*).

In the Supp Fig 4 legend, it is stated that data were fitted with a two-step sequential binding model, but this is only true for the NTD+SL4ext data, not the NDT+SL4 data that were fitted with a single binding site model in the revised version of the manuscript. This should be corrected in the figure legend.

This is indeed correct and we highly appreciate the careful reading. We have now changed fitting of all ITC curves (which are new, see comment below) to a consistent model, and the legend should state the right information.

The stirring speed of 500 rpm is quite unusual for VP-ITC (usually 310 rpm; higher rate desirable for experiments where aggregates are formed; was this the case? Could authors comment on this?).

In fact, the default stirring speed according to the software is 750 rpm. Initially, we have reduced this value to 500 rpm as this is a value that we have been successfully using in prior experiments. However, following the reviewer's suggestions (and since we indeed do not have issues with aggregate formation), we adjusted the stirring speed to 310 rpm for the ITC experiments we performed for this resubmission (see also comments below).

In addition to these details, I'm still highly skeptical about ITC data and interpretation. First, there is no mention in the ITC method section of blank experiments and of a negative control experiment with an unrelated RNA of similar size. These experiments seem to me crucial insofar as the experimental ITC data are really very weak (low signal/noise and extremely weak enthalpy) and the obtained K_d for SL4 is more or less within the limits of what can be measured in biomicrocalorimetry: according to the expected pI of the protein (around 9.6), there is necessarily an affinity for any (negatively-charged) RNA which is not negligible due to non-specific electrostatic interactions, especially since the experimental conditions have been modified to be relatively low in salt (50mM KCl only). This should be clarified.

We thank the reviewer for the constructive suggestion to increase the impact of our ITC data. We have undertaken the following add-ons: We have now re-measured interactions of NTD with SL4 and SL4ext and used an inversed titration, i.e. titrating NTD into RNAs, which has facilitated a better data analysis and yielded data of higher quality. We have also included SL1_ext^{SL1} as a control RNA, which comprises a stem loop and an ssRNA part, similar to SL4ext. We had included this RNA to support our claim of specificity in the stem-loop-ssRNA combination of SL4ext (see NMR data). As expected by the reviewer and in line with NMR, we do see weak, nonspecific binding at low-salt conditions, yet weaker than for SL4.

We have then probed the three RNAs in NTD-binding under conditions of 250 mM salt. In line with our data from NMR, we do not further see interactions of SL4 and SL1_ext^{SL1} with NTD, while one binding event is still given at low μ M-affinity for SL4ext (more driven by entropy).

As correctly suggested, we have now also included blank runs of NTD into buffers and buffers into SL4 RNAs, which show no impactful heat production with relevance for the fit of bindings. Subtraction of blanks did not alter the fitting outcome of data.

We hope, with this set of valuable experiments, to convince the reviewer of our ITC data. We have adjusted the main text (lines 141-143, 270-276, 379-381, 421), the Suppl. Fig. 4, Supplementary Methods, and have updated the Source Data file.

Second, about ITC data for NTD-SL4 interaction, the authors processed these data with a one-site model (I wouldn't, however, claim – as stated in the response to reviewers - that the expected 1:1 binding mode is clear!), thus excluding any unspecific interactions between the protein and the RNA, which looks quite unexpected according to the high pI for the protein and the low salt conditions.

This is indeed a very good point and we may have misunderstood this comment before. For consistency, and as given in the Methods section, we have now used an asymmetric sequential binding model in all ITC runs to allow equivalent bindings to SL4ext (and with n unequal 1 if given). Interestingly, we do see a 1:1 binding of NTD to SL4 and SL1_ext^{SL1}, while additional interactions are too weak to reveal meaningful fittable data. Again, this in line with NMR, where higher stoichiometries would lead to significantly broader lines for SL4 than the ones we find. Possibly, multiple unspecific interactions at a time might be disfavored because of steric reasons, while we do not exclude them for yet larger RNAs.

As stated above (see previous point) increased salt concentrations indeed abolish those (charge-driven) interactions, while specific interaction with SL4ext (at the Ext site) remains measurable even at 250 mM salt.

Third, about the ITC data for NTD-SL4ext interaction, I do not understand the arguments put forward by the authors to explain the model used and the observed stoichiometries (“...as the two binding events are not completely distinct we assume a limitation in resolution of ITC for fitting. This gets obvious from the non-intuitive molar ratios of ~0.5 and 1.2, likely caused by the sum up of binding curves...”). If a correct model is used for ITC data fitting, stoichiometries derived from the model should be correct, in agreement with the model, and no limitation of resolution limit applies. This can possibly be tested using a reverse titration and a global fit. Again, with this SL4ext RNA fragment it would be surprising to observe no signal from a non-specific interaction given the protein pI and the salt conditions.

As summarized above, we have gratefully followed the suggestion of inverse titration, which yielded data of higher quality, but as such still in accordance with the original ITC outcome. For SL4ext, we have now obtained n values of 0.8 and 1.4. We agree that the lower-affine, nonspecific binding at an $K_D > 60 \mu\text{M}$ (which is also reflected in SL4 and SL1_ext^{SL1}) is per se at the limit of ITC suitability in our setting. Nevertheless, the fitting of this weak binding was consistently found independent of the model (A+B, A+B+B symmetric or asymmetric etc.), such that this only fittable event is a reliable number, also seeing replicates. We also agree that the NTD would reveal such binding to any arbitrary RNA under the low-salt condition, which is in line with its promiscuous RNA-binding role and supported by EMSA and NMR, while only few RNA motifs like Ext lead to a significantly enhanced complex stability.

Reviewer #3:

I appreciate the time and effort that the authors put in in addressing the comments from the last draft. However, interpretation of the SAXS data is still unclear. Here are my questions and concerns.

1. In the last version, there was clear evidence of a problem at low q. The authors reply that this is potentially due to radiation damage, which should be concerning. If the sample is damaged by the beam, how can they trust the data? Usually most experiments begin by finding conditions that avoid radiation damage, by reducing exposure or adding attenuation.

We agree and do see that our data recorded in the previously measured phosphate buffer conditions may suffer from radiation damage leading to unreliable data analysis and interpretation. We have now undertaken the following adjustments: i) We have included an expert as co-author to increase the quality of our data based on discussions about possible buffer conditions – as close to physiological as possible – and crucial components to reduce radiation damage in SAXS measurements. ii) We have now recorded SAXS data of individual components again in HEPES, NaCl, TCEP and sodium nitrate. iii) We have also reprocessed the original SAXS data in phosphate, which has led to improved data quality (including the low-q region) (see Supplementary Fig. 8). Of note, the potential radiation damage mentioned in the initial text, was found to be a consequence of likely beamline instabilities/parasitic scattering, and the inability of automated routines in CHROMIXS to identify these artefacts. Careful reevaluation and manual reprocessing of the data has resolved this issue, and the approach

through which to overcome beamline-artefacts to generate the reprocessed background-corrected SAXS data is now described in the (Supplemental) Methods.

We would like to mention, that the new HEPES buffer is less optimal for the biological activity of viral RNP formation and thus may be looked at with caution (well discussed before by Jeffries et al., 2015, J Synchr. Rad.), esp. with respect to additives like TCEP and sodium nitrate that would affect both the highly charged NTD and RNA sample states! We have, however, included a direct comparison of raw scattering curves for NTD and RNAs, and the derivable parameters like R_g and D_{max} (Supplementary Fig. 8f, Source Data). The basically identical outcome shows that data recorded in the original phosphate buffer (the physiological condition, where all other experiments have been made in) still represent the monomeric states and integrity of all used components irrespective of possible radiation damage, as one can transfer findings from HEPES back to phosphate. Additionally, we have checked base-pairing (and thus secondary structure) for the RNAs by 1D-NMR in HEPES, indicating no buffer induced changes (not included here).

Of note, the HEPES buffer conditions allow for high-quality data, e.g. well-reflected in the model fit of NTD with the NMR ensemble, which has again improved. All components have also been subjected to parallel MALS, which once more confirms their monomeric, non-aggregated state. We thus thank the reviewer for the constructive suggestion of rethinking SAXS experimental conditions. For consistency with all other experiments, we would still provide all SAXS data in phosphate in the main text, while SAXS data measured in HEPES buffer are shown in the Suppl. Fig. 8f (including a direct comparison to phosphate) and the Source Data table. We have also deposited the new SAXS data in HEPES to the SASBDB.

2. It is hard to piece together exactly what was the experimental procedure. Was the aSEC done in the lab, prior to shipping the sample? Hence, days in advance of the beamtime? The SEC-SAXS shows that the sample refractionates on the beamline. How do you know that you picked the right fraction to look at? It seems that the largest fraction was selected, but is there another way to be sure that this is the correct one? Is the frame number reproducible between runs? The data are plotted in a way that makes it very difficult for a reader to assess (and this should be something that the authors report on). Please justify how the proper fractions were determined.

We apologize if the experimental procedure of aSEC versus SEC-SAXS measurements was not clearly enough delineated. Certainly, this should be comprehensible for the reader. For the first round of SEC-SAXS in phosphate, all samples were mixed in our lab from purified components without another SEC run on-site. Samples were shock-frozen in liquid nitrogen and delivered to the beamline, defrozen and subjected to SEC-SAXS after extensive centrifugation. We had to switch to this unified procedure at the beginning of Covid-19, while in principle they -by default- would have been mixed directly there from non-frozen NTD and RNA. We do know that shock-freezing and thawing does not negatively influence sample states of the NTD, and not at all RNAs, as this is the default type of long-term storage. In fact, we used transport on dry ice to make sure maximal possible control over sample state/temperature. Samples were normally delivered 1-2 days before beamtime and locally stored at -80°C as by agreement.

For the new round of SEC-SAXS in HEPES buffer, we were able to return to in-person measurements and all samples were transported on ice (the day before), mixed on site, centrifuged, and subsequently loaded (as described in the Supplementary Methods section).

SEC-SAXS runs provide scattering over frames as default readout. Note that the SEC run is carried out at room temperature (as we also did in house for the aSEC analysis). For HEPES

runs of individual components, there is clear proof for the “right” frames, which have been combined with MALS and now provide a MW-based readout within the selected peaks/frames.

For the comparison in complex formation, peak fraction selection used the complete number of frames forming the central peak region and showing sufficient signal-to-noise for meaningful exploitation, and importantly consistency in the R_g correlation through the selected peak frames (stable R_g through the sample SEC peak, ± 0.1 nm). Details/modifications for individual samples are described in the Methods section. Note that the frame number is a rough parameter for comparison as it depends on the flowrate, which may vary for technical reasons. However, in line with our aSEC analysis, stable RNP complexes in SEC-SAXS can be correlated with lower frame numbers compared to the respective free components (although not precisely transferable to exact elution volumes). This is visualized and included in Suppl. Fig. 8g.

We have adjusted the respective parts in the Methods sections and hope to be comprehensible now.

3. In addition, the presentation of curves in Supp Figure 8 is quite confusing. Why are the x axes different for the SAXS profiles? It is quite hard to compare the curves. Furthermore, the points are so light and hard to see, that it is hard to agree with the statement that their fit is ‘excellent’. I see significant deviations from the pdb generated curve, and I cannot even resolve the error bars.

We have improved the graphical quality of this figure now to provide optically resolvable items, data points and errors. This also includes the homogenization of x-axes. (Note that the previous way of plotting them was based on the default angular unit $1/\text{nm}$ in scattering curves over the full angular range (without fit), while all curves in overlay with PDB-derived theoretical curves from Crysol are shown in the fitted range in $1/\text{\AA}$ (default output)). We have adjusted the units in the Suppl. Fig. 8 for homogeneity and included the full angular space (0.7 \AA^{-1}) for experimental curves in all plots now. For Crysol fits, we consistently used the angular space until 0.4 \AA^{-1} for the NTD and 0.5 \AA^{-1} for the RNAs. Consistent units are also provided in the main text SAXS Fig. 4.

4. The radius of gyration values from Table 2 cannot possibly be real.

We absolutely agree. We here had not shown the proper unit (nm instead of \AA). We thank the reviewer for pointing us at this mistake and have now, in the process of working over all data, corrected the main Table 2 as well as the new Suppl. Table 2 accordingly, and the same for the newly acquired data in HEPES buffer (see Source Data file).

5. Assuming that the SAXS data are OK (which they might be, but I cannot be confident, given the concerns raised above), the authors MUST provide compelling argument that the Porod volume for a protein RNA complex can be interpreted as they do in the manuscript. When two molecules associate, it is quite reasonable to expect an increase in the Porod volume, so this seems fine; however, I am concerned that they are overinterpreting the changes as a remodeling of the RNA. Specifically, they are looking at the interaction of an RNA with a protein. These molecules have different electron densities, so I would have to very carefully consider how these different densities affect the computation of the Porod volume. If there were two proteins associating, the molecular density in the $I(0)$ would cancel out the term in the integral, but it is certainly not obvious to me that this same cancellation occurs for two interacting molecules with different electron densities. If this has been

worked out (which it may well have been), the authors need to provide a reference describing the calculation. They might work with a SAXS expert to do the calculation otherwise.

We are thankful for this correct comment; we have taken a too conservative approach when interpreting the Porod volume as a single parameter in isolation. Indeed, the scattering contributions from the different regions of scattering length density, in addition to cross-term contributions from between these regions within the RNPs, complicates any interpretation of the Porod volume in and of itself with respect to RNA remodeling (although not the actual magnitudes of the Porod volume).

The initial idea was to simply demonstrate that the engagement of NTD with the RNAs either yields isolatable complexes, or as is the case with SL4, a system that is more unstable and tends toward disassociation, visible under the influence of SEC. Consequently, we have decided to modify the SAXS summary figure and tables towards a clearer output, reporting differences in the $P(r)$, in combination with the R_g , changes in R_g , Porod volumes and MW estimates in relation to the RNA/NTD interaction(s). In addition, the comparison of the scattering profiles of the individual RNAs and respective NTD-RNA samples are now also shown in the main text, i.e. in the revised Fig. 4. When taken together, the results show that those RNAs containing the Ext element (Ext itself, or SL4ext) undergo a huge change in R_g , of almost 0.7 nm compared to the respective RNA components alone, (while maintaining estimated 1:1 binding). The SL23-NTD undergoes a moderate 0.23 nm increase in R_g relative to SL4 alone, and the SL4 system produces a quite miserable 0.13 nm increase that is indicative of the scattering being dominated by the SL4 RNA, and the lack of stable complex formation. We thus now use SAXS to provide another proof for complex formation as read by overall geometries from SEC-SAXS data, which has been a valuable tool in similar studies before. As such, we rephrase the motivation and focus on statements on selectively formed RNPs along the lines of following in the main text: "This may suggest differences in the modes of NTD/RNA interactions, e.g., conformational rearrangements of the RNA on complex formation, or a different positioning of the NTD relative to the center of mass of the corresponding RNA within the complex." (lines 237-240)

We hope that the modifications meet the justified concerns of this reviewer regarding the Porod volumes, in that we now draw on a more holistic approach to the SAXS data analysis to include the reporting of all structural parameters, and interpret the results in context of selective modes of interaction between NTD and RNA.

To this end, we have comprehensively adjusted the main text SAXS section on pages 7/8.

6. Additionally, the data show some artifacts at the low q values, both the increases discussed above, and now some downturns that may be the result of interparticle interference effects. Since the $I(0)$ value is so critical to the computation of Porod volume, it must be properly obtained, and the lack of acknowledgement of this, suggests that the authors might underestimate the difficulty of the analysis they are proposing. This is definitely a non-trivial experiment.

Please see the comments above.

7. Please carefully proofread the table of SAXS parameters. There are some errors. The author is Trehwella, not Trehwelly.

We apologize for this typo. It is now corrected. We have also gone through all entries of the SAXS parameters table(s) and indeed adjusted incorrect parameters/numbers. It should now all be correct.

REVIEWER COMMENTS

Reviewer #1 (Remarks to the Author):

In this new revised version of the manuscript the authors again significantly improved the manuscript and addressed most concerns. Nevertheless, despite the notable efforts of the authors, ITC data analysis is still not satisfactory for the following reasons:

- minor detail: in the Supplementary Mat&Met, ITC section: "However, at normal salt conditions only a single...". What is "normal salt conditions"? 50mM (low) and 250mM (high) KCl have been tested. Not sure that 50mM salt can be considered as normal compared to physiological conditions...

- major issue: in this version of the manuscript, a reverse ITC titration is presented (protein into RNA) but the previous experiments are no more present (RNA into protein). It is extremely unfortunate that the two experiments (titration + reverse titration) are not presented in order to make a global fitting of the data, which is much more robust (and available with the software suite used by authors) than a single fit.

- about the fitting of the data (major issue): I'm satisfied about data processing in 50mM salt conditions (although stoichiometry, DeltaS and DeltaG are missing, only values for DeltaH and Kd are presented). It's nice to see the lack of interaction (unspecific interaction) with SL4 and SL1_extSL1 in high salt conditions. BUT, data collected in high salt with SL4ext (supp fig 4b) clearly show a complex binding mode and is clearly not a single binding site as used by authors and as stated in the text of the Supplementary Material! By extracting the data from the authors and using a model with two independent sites, one can get an absolutely perfect fit (see attached picture), with an entropy-driven tight binding + a second enthalpy-driven weak binding (and an overall active fraction of about 0.3). It remains to be seen whether the model is justified, but it clearly shows that the simple 1:1 model used by the authors cannot be applied in these conditions. This needs to be clarified and a global fitting will probably help.

- about legend of supp fig 4b (minor issue): "while SL4ext still provides affinity for the first binding site, which is driven by entropy (in line with NMR titrations at high salt...)". The fit precisely shows that enthalpy also positively contributes to the interaction-driven interaction, so it is rather an entropy- and enthalpy-driven interaction. But anyway, the fit is clearly not correct, as discussed previously.

Reviewer #3 (Remarks to the Author):

The presentation of the SAXS data is much improved over the last version, specifically the data shown and the improved clarity of interpretation. My only remaining concern is with the profiles acquired on SL4 and SL4 plus the NTD. The $P(r)$ curve with and without the NTD does not look very different. I assume this is because the NTD signature is not easily resolved from the SL4 signature, or perhaps the fraction that was selected corresponds to the SL4, which does not bind to the NTD.

I ask that the authors clarify this point in the manuscript.

We thank both reviewers for appreciating our efforts to improve the manuscript. We here hope to sufficiently address the remaining open concerns.

We have marked edits of the third round with blue color in the tracked-changes-versions of manuscript files with respect to the previous, second revised version.

Find below the detailed responses to reviewers 1 and 3.

Reviewer #1:

In this new revised version of the manuscript the authors again significantly improved the manuscript and addressed most concerns. Nevertheless, despite the notable efforts of the authors, ITC data analysis is still not satisfactory for the following reasons:

- minor detail: in the Supplementary Mat&Met, ITC section: “However, at normal salt conditions only a single...”. What is “normal salt conditions”? 50mM (low) and 250mM (high) KCl have been tested. Not sure that 50mM salt can be considered as normal compared to physiological conditions...

This is absolutely correct and we have corrected this mistake throughout the texts. We highly appreciate this important comment.

- major issue: in this version of the manuscript, a reverse ITC titration is presented (protein into RNA) but the previous experiments are no more present (RNA into protein). It is extremely unfortunate that the two experiments (titration + reverse titration) are not presented in order to make a global fitting of the data, which is much more robust (and available with the software suite used by authors) than a single fit.

We apologize if this was a misunderstanding in the previous revision. While this is a more than understandable comment and suggestion, we would like to underline that the original (‘forward’ titrated) ITC data have not been recorded with the same systematics and, as we have confessed, have been of clearly lower quality, at least for SL4. Still, as in principle they are in line with the current ITC and the other binding data, we have re-included the forward ITC comparison in Suppl. Figure 4c and the Source Data for completeness. To this end, we have also thoroughly reanalyzed them and now provide much better fits for NTD when titrated with SL4 and SL4ext. We find that for both RNAs the two types of titrations corroborate each other very well in the obtainable thermodynamic parameters, affinities and n-values. Finally, in Suppl. Figure 4d we now provide affinities derived for NTD binding to SL4 and SL4ext, which are derived from all accomplished ITC measurements at (comparable) low-salt conditions. We now refer to those integrated K_d values at their first mention in the first results section on page 5, while we explicitly mention reverse-ITC-derived affinities in the low-salt vs. high-salt comparison. We hope this approach satisfies the reviewer’s concerns.

We would like to point out that all ITC data per se well support the findings from the other methods, i.e. NMR, EMSA, and also aSEC. We are aware of the complex options of NTD binding to large, folded but dynamic RNAs and the challenges those systems pose to ITC. We would like to underline that the focus was initially to be set on the comparison between the specifically vs. non-specifically bound elements, which we think is clearly backed-up with the ITC data. We confess that, during the various rounds of revisions, we have improved ITC data

quality and fitting correctness/outcome. We though hope that the current depiction of ITC now both represents data and their analysis of sufficient quality AND reflects or supports NTD-RNA binding models as suggested by the manifold other data in our manuscript, in particular regarding stoichiometries.

- about the fitting of the data (major issue): I'm satisfied about data processing in 50mM salt conditions (although stoichiometry, DeltaS and DeltaG are missing, only values for DeltaH and Kd are presented). It's nice to see the lack of interaction (unspecific interaction) with SL4 and SL1_extSL1 in high salt conditions.

We hope not to misunderstand the reviewer. Please note that all thermodynamic measures incl. DeltaS and DeltaG as well as stoichiometries were and are given in the SI figure and the Source Data for all presented ITC data. Apart from that, we are glad about the reviewer's appreciation of our newly integrated data.

BUT, data collected in high salt with SL4ext (supp fig 4b) clearly show a complex binding mode and is clearly not a single binding site as used by authors and as stated in the text of the Supplementary Material! By extracting the data from the authors and using a model with two independent sites, one can get an absolutely perfect fit (see attached picture), with an entropy-driven tight binding + a second enthalpy-driven weak binding (and an overall active fraction of about 0.3). It remains to be seen whether the model is justified, but it clearly shows that the simple 1:1 model used by the authors cannot be applied in these conditions. This needs to be clarified and a global fitting will probably help.

We agree that a more comprehensive fit, leading to the provided plot (we highly appreciate it) of the high-salt titration with SL4ext is possible. We highly appreciate this picture and need to say that with our initial fit settings we did not get to this very fit curve in first instance. We now have undertaken this exact procedure as shown in the Suppl. Figure 4B and the Source Data. Indeed, as shown in the accompanying bar chart, we do find the mentioned entropy-driven first binding and an enthalpy-driven second one. We now adjusted this in the legend and made a short addition/adjustment to the main text and MatMet. While we cannot fully integrate the two derivable numbers into a conclusive model, esp. regarding the other methods' data, the somewhat lowered affinity is well in line with the faster exchange regime seen by NMR at high salt. However, the engagement of RNA with two NTDs in two possible ways at 250mM salt may not be fully resolved by NMR (likely revealing a mix of effects). As the major conclusion, we would like to remain with the fact that high salt retains affine binding of the more specific SL4ext vs. SL4 or SL1, while the possible modes of interaction and less-specific contributions

remain complicated to interpret without further (structural) data. We thus hope to satisfy the reviewer's concerns about the precise fitting and (un)specific binding contributions.

- about legend of supp fig 4b (minor issue): "while SL4ext still provides affinity for the first binding site, which is driven by entropy (in line with NMR titrations at high salt...". The fit precisely shows that enthalpy also positively contributes to the interaction-driven interaction, so it is rather an entropy- and enthalpy-driven interaction. But anyway, the fit is clearly not correct, as discussed previously.

As provided in our reply to the previous comment we have improved the fit and now adjusted the text in the legend accordingly.

Reviewer #3:

The presentation of the SAXS data is much improved over the last version, specifically the data shown and the improved clarity of interpretation. My only remaining concern is with the profiles acquired on SL4 and SL4 plus the NTD. The $P(r)$ curve with and without the NTD does not look very different. I assume this is because the NTD signature is not easily resolved from the SL4 signature, or perhaps the fraction that was selected corresponds to the SL4, which does not bind to the NTD.

I ask that the authors clarify this point in the manuscript.

We thank this reviewer for appreciating our efforts to adjust the analysis and presentation of SAXS data, and also for giving us the opportunity to clarify this important, technical detail again. Indeed, this observation exactly underlines the core experimental outcome, in which the selected SEC fractions for both runs, i.e. SL4 alone and SL4 together with NTD, (almost) exclusively represent SL4 alone, because no complex formation can be observed for SL4. As we have tried to use fractions that would not overlap with free NTD (as good as possible, in order to distinguish real complexes from the co-presence of non-interacting species), one would in fact expect almost identical scattering curves and $P(r)$ profiles. As SL4 and NTD do run very close in SEC (see also Suppl. Figure 8g for the R_g traces) a full separation is hard to achieve, but clearly no quantitative complex formation is observed. Suppl. Figure 8g also shows the almost complete superimposition of apo and NTD-SL4 sample frames in SEC-SAXS and via their R_g .

To clarify again the context of this observation, we have now complemented the present sentence on page 8 of the main text: "In contrast, it appears that for SL4 the scattering of both samples is exclusively caused by the RNA component, where both the SL4 and the NTD-SL4 SAXS profiles appear near-identical and generate very similar structural parameters and $P(r)$. This underlines the sole presence of non-bound RNA in the respective fractions even in the presence of NTD (Supplementary Fig. 8g) and is best seen by the lack of increase in MW in the presence of NTD, different from the other three RNAs (Fig. 4a)."